# On Hamming–Lipschitz Type Stability of the Subdominant (Minmax) Ultrametric: Theory and Simple Proofs

**Alokendu Mazumder**[*]                                    *alokendum@iisc.ac.in*
*Department of Cyber Physical Systems*
*Indian Institute of Science, Bengaluru*

**Arnab Roy**[†*]                                           *arnabroy@iisc.ac.in*
*Department of Computer Science and Automation*
*Indian Institute of Science, Bengaluru.*

**Punit Rathore**                                           *prathore@iisc.ac.in*
*Department of Cyber Physical Systems*
*Centre for Infrastructure, Sustainable Transportation and Urban Planning*
*Indian Institute of Science, Bengaluru*

**Reviewed on OpenReview:** *https://openreview.net/forum?id=R4ASOCp3uM*

## Abstract

The subdominant (minmax) ultrametric is a canonical tree-structured summary of a dissimilarity matrix, arising equivalently as the ultrametric induced by single-linkage clustering. While its classical stability theory is usually formulated in $\ell_\infty$ or Gromov–Hausdorff terms, such bounds are poorly suited to sparse perturbations that alter only a few pairwise distances. We develop an $\ell_0$-type stability theory for this operator. Our analysis shows that sparse edits propagate only through the *minimum spanning tree* (MST): a pairwise ultrametric value can change only if its tree path crosses an edited edge or a cut newly exposed by an edited off-tree edge. This yields a sharp per-edit exposed-cut score and a tree-only global envelope, leading to Hamming–Lipschitz bounds on the number of ultrametric entries that can change. We also prove sharpness results showing that this dependence on tree geometry is unavoidable: under strict cut separation, the tree-edge bound is attained exactly, and for off-tree edits, there are explicit families in which one edited distance changes $\Theta(n^2)$ ultrametric entries. In addition, we prove a conditional near-additivity principle for multiple edits under certified large per-edit changed regions and negligible aggregate overlap. Experiments on deep-embedding graphs show that the resulting structural scores provide useful vulnerability diagnostics for hierarchical representations.

## 1 Introduction

Hierarchical clustering (Ward Jr, 1963) provides a fundamental way to represent relational data through nested partitions and dendrograms (Shepard, 1962). Among all possible hierarchies, the subdominant (or minmax) ultrametric (Sibson, 1971; Hartigan, 1985; Jain & Dubes, 1988) occupies a canonical position: it is the subdominant ultrametric that is maximally dominated by the given dissimilarity and plays a canonical role in hierarchical clustering and metric geometry. Formally, given a metric space $(X, d)$, the minmax ultrametric can be viewed as the output of an operator $U : (X, d) \mapsto (X, u_d)$, where $u_d : X \times X \mapsto \mathbb{R}$ is the minmax ultrametric associated with $d$. This operator maps a dissimilarity to its associated subdominant ultrametric, equivalently the merge-height function of single-linkage clustering. Carlsson et al. (2010) established that the dendrogram of *single linkage clustering* (SLC) and the minmax ultrametric are equivalent representations

---

[*]denotes equal contribution. [†] Work done at VISTA Lab, Department of Cyber Physical Systems, IISc Bengaluru

of the same hierarchical structure. This quantity coincides exactly with the merge height of the two points in the single-linkage dendrogram. Hence, single-linkage hierarchical clustering can be viewed as computing the *maximal ultrametric dominated by the original distances*, providing a precise geometric correspondence between dendrograms and ultrametrics. Further, Carlsson et al. (2010) formulated a rigorous mathematical framework for hierarchical clustering by identifying it with a mapping from finite metric spaces to ultrametric spaces. Their key theoretical result is that the minmax ultrametric map is 1-Lipschitz (non-expansive) with respect to Gromov-Hausdorff ($\mathcal{GH}$) metric. Thus, the minmax ultrametric (dendrogram of SLC) is non-expansive under arbitrary perturbations of the input metric in $\mathcal{GH}$ sense, implying that small changes in distances cannot amplify in the induced ultrametric. In contrast, other linkage-based operators such as complete or average linkage, fail to satisfy this non-expansive property. Their stability theorem establishes single linkage as the unique hierarchically consistent and Lipschitz stable ultrametric projection. Regarding assumptions on perturbations, their analysis is fully general; no probabilistic or noise model is imposed. The only requirement is that the metric perturbation be bounded in the Gromov–Hausdorff sense, meaning that all pairwise distances between the two metric spaces differ by at most a small additive $\varepsilon$. Under this assumption, every ultrametric distance changes by at most $\varepsilon$. Thus, their stability theorem captures uniform, global perturbations of the metric, but does not address sparse or localized adversarial edits.

Chowdhury et al. (2016) further analyzed stability in more concrete norms. They prove that the minmax ultrametric operator $U : (X, d) \mapsto (X, u_d)$ is 1-Lipschitz under the sup norm. Mathematically, $\| u_d - u_{\tilde{d}} \|_\infty \leq \| d - \tilde{d} \|_\infty$, for all metrics $d, \tilde{d}$ on $X$. Crucially, they formalized a duality between Gromov's tree embedding and the ultrametric structure produced by SLC. They introduced a measure of deviation from ultrametricity, quantifying how far a finite metric space is from being perfectly treelike. Through this duality, they proved that the single-linkage dendrogram computes the ultrametric that minimizes additive distortion up to a bound depending on the space's ultrametricity and doubling dimension. In essence, the single-linkage dendrogram corresponds to an optimal ultrametric tree embedding whose distortion reflects both the local *ultrametricity* of the data and its intrinsic dimensional complexity. Together, these results position the single-linkage dendrogram as both an optimal low-distortion tree embedding and a globally stable (Lipschitz-continuous) map from metric data to hierarchical structure.

Recently, Mikhailov (2025) extended the stability result of Carlsson et al. (2010) to the full Gromov–Hausdorff class of all (possibly unbounded) metric spaces. He showed that the canonical subdominant (min–max) ultrametricization mapping $U : (X, d) \mapsto (X, u_d)$, obtained from the Carlsson–Memoli construction, is *1-Lipschitz* with respect to the Gromov–Hausdorff distance, not only on bounded spaces but on arbitrary metric spaces. This viewpoint interprets $U$ as a non-expansive map between *clouds*, i.e., equivalence classes of spaces at finite Gromov–Hausdorff distance. Moreover, for any dotted connected metric space $A$, he exhibited an inverse relationship between $U$ and Cartesian products with $A$: on the cloud of bounded ultrametric spaces, the map $\Psi : X \mapsto X \times A$ is an isometric embedding and $d_{\mathcal{GH}}\big(U(\Psi(X)), X\big) = 0$, so that $U(\Psi(X))$ is (Gromov–Hausdorff) isometric to $X$, while ultrametric spaces are precisely the fixed points of $U$. Conceptually, this places the ultrametricization operator as a globally Lipschitz-stable transformation over the Gromov–Hausdorff landscape, further linking metric geometry with hierarchical clustering.

## 1.1 Gaps in Previous Theory and Motivation

Existing stability results for the minmax ultrametric $u_d$ address *uniform* perturbations, proving non-expansiveness in the entrywise $\ell_\infty$ metric, and via standard comparisons, in the Gromov–Hausdorff framework. These guarantees bound the *magnitude* of change everywhere but are agnostic to the *sparsity* and *locality* of the perturbation: a single large edit renders $\|d - \tilde{d}\|_\infty$ large, and the resulting bound allows every entry of $u_d$ to move by that amount, even when the true effect is confined to a tiny portion of pairs. In particular, the uniform theory provides no control over the *extent* (support size) of the induced change in $u_d$ when perturbations are sparse.

We close this gap by analyzing the ultrametric map $U : (X, d) \mapsto (X, u_d)$ in a Hamming-type setting on pairwise dissimilarity matrices. Concretely, for a finite metric $d$ on $X$, we encode $d$ as its upper-triangular distance vector and equip this space with the Hamming metric

$$d_H(d, \tilde{d}) \;=\; \#\big\{\{x, y\} \subseteq X : d(x, y) \neq \tilde{d}(x, y)\big\}, \tag{1}$$

i.e., the number of pairs whose dissimilarities are edited. Equivalently, $d_H(d, \tilde{d}) = \|d - \tilde{d}\|_0$, where $\| \cdot \|_0$ counts nonzero coordinates. Although $\| \cdot \|_0$ is not a norm, it induces the bonafide Hamming metric $d_H$, and we establish a sparsity-sensitive $\ell_0$-type theory in which the changed-pair set is controlled by per-edit exposed-cut regions. In particular, the analysis yields a tree-only global envelope $\bar{L}_T$, giving bounds of the form $\|u_d - u_{\tilde{d}}\|_0 \leq \bar{L}_T \|d - \tilde{d}\|_0$. $\bar{L}_T$ depends only on the MST structure; in the worst case on an $n$-node graph, $\bar{L}_T \leqslant \binom{n}{2}$. Thus, our result complements the classical $\ell_\infty / \mathcal{GH}$ non-expansiveness along an orthogonal axis: the prior theory controls *how much* entries may move under uniform noise, while our Hamming-metric guarantee controls *how many* entries can change under sparse edits. Together, these yield a magnitude-versus-extent stability picture that was previously unavailable for the ultrametric operator.

| Continuity (Lipschitz) Type | Noise Model / Assumptions | Lipschitz expression | Reference |
|---|---|---|---|
| $\ell_\infty$-continuity (sup-norm) | Bounded entrywise perturbations of all pairs. | $\|u_d - u_{\tilde{d}}\|_\infty \leqslant \|d - \tilde{d}\|_\infty$ | Carlsson et al. (2010); Dey et al. (2017) |
| $\mathcal{GH}$ stability | Arbitrary perturbation measured in $\mathcal{GH}$-distance. | $d_{\mathcal{GH}}(u_d, u_{\tilde{d}}) \leqslant d_{\mathcal{GH}}(d, \tilde{d})$ | Carlsson et al. (2010) |
| $\mathcal{GH}$ (semi-)stability | Stable in $\mathcal{GH}$ only when the input metric is (nearly) ultrametric. | $d_{\mathcal{GH}}(u_d, u_{\tilde{d}}) \leqslant d_{\mathcal{GH}}(d, \tilde{d})$ | Martínez-Pérez (2015) |
| $\ell_\infty$ (restricted perturbation) | Additive perturbation on a subset $S'$(or insertion/removal of points). | $\|u_d - u_d^{S'}\|_\infty \leqslant \|d - d^{S'}\|_\infty$ | Chowdhury et al. (2016) |
| $\mathcal{GH}$ stability for unbounded spaces | Entrywise-bounded perturbations for not-necessarily bounded metric spaces. | $d_{\mathcal{GH}}(u_d, u_{\tilde{d}}) \leq d_{\mathcal{GH}}(d, \tilde{d})$ | Mikhailov (2025) |
| $\ell_0$ (pair-count; Hamming metric) | Adversarial *sparse* edits of up to $k$ pairs (no magnitude bound); propagation constrained by MST structure. | $\|u_d - u_{\tilde{d}}\|_0 \leq \bar{L}_T \|d - \tilde{d}\|_0$ | This work |

Table 1: Stability regimes for the minimax/subdominant ultrametric. Classical uniform bounds control *magnitude* ($\ell_\infty$/GH), while our Hamming-space ($\ell_0$) bound controls the *extent* of change under sparse edits.

## 1.2 Minmax Ultrametrics in Modern Machine Learning

Zhu et al. (2017) noted that among common hierarchical clustering methods, only single-linkage (the minmax ultrametric) is stable under small perturbations of the input weights (dissimilarities) and consistent in the infinite-sample limit. Specifically, they prove that SLC is the sole method satisfying: *if the number of i.i.d. sample points goes to infinity, the output ultrametric converges (a.s., in Gromov-Hausdorff sense) to the true multiscale structure of the data distribution's support.* Dey et al. (2017) study temporal hierarchical clustering by fitting each time slice with an ultrametric and enforcing small inter-time distortions. They show that the generic $\ell_\infty$ nearest-ultrametric fit can be unstable under metric perturbations, and therefore replace it with the *minmax ultrametric* $u_d$, defined as the maximum edge weight along the MST path; $u_d$ is the unique $\ell_\infty$-closest ultrametric that does not increase any input distance. This choice yields temporally coherent single-linkage dendrograms (since $u_d$ is the single-linkage/minmax ultrametric) while restoring stability in their temporal objective.

Devijver et al. (2024) studied the stability in high-dimensional network inference by inserting a single-linkage hierarchical clustering step before graphical lasso, and proves that the resulting dendrogram, hence the minmax ultrametric underlying single linkage is stable under data perturbations, unlike average linkage. Concretely, the classical two-step decomposition first clusters variables via single linkage on a similarity derived from absolute sample covariances, then fits graphical lasso inside the resulting modules; prior work shows this Step-1 clustering is exactly SLC on that similarity, making the dendrogram the algorithm's

structural backbone. The authors provide theoretical bounds controlling distances between dendrograms built from two samples and show in simulations and real data that single linkage based modules are markedly more stable than alternatives, while complete/average linkage can be unstable. Overall, the paper is a recent, theory-driven application where the minmax ultrametric (via single linkage/MST path-max) is explicitly used to stabilize hierarchical decomposition before sparse graphical model estimation.

Recent methods leverage ultrametrics as algorithmic backbones for broader clustering objectives and pipelines, e.g., showing that center-based objectives can be solved optimally on ultrametrics and producing rich cluster hierarchies (Draganov et al., 2025), and, in density-based settings, building MST/minmax style hierarchies whose slices enjoy formal stability/consistency (Rolle & Scoccola, 2024; Ritzert et al., 2025).

Beyond classical hierarchical clustering, recent work formulates ultrametric fitting as an optimization problem amenable to gradient-based learning and end-to-end training. Learning an ultrametric therefore, amounts to inducing a hierarchy from data rather than merely running a procedural agglomeration. Chierchia & Perret (2019) proposed a continuous optimization framework for learning ultrametrics: they replace the ultrametric constraint by a minmax formulation so one can optimize over ultrametric matrices with standard gradients. Their objective flexibly combines closest ultrametric fidelity with task-driven terms (e.g., Dasgupta's HC objective, cluster-size regularization, triplet constraints), and scales to large graphs with performance comparable to strong agglomerative baselines. In a related vein, Chen et al. (2024) cast tree–Wasserstein regression as ultrametric learning: they learn a tree metric (ultrametric) so that the induced tree–Wasserstein distance approximates the underlying *optimal transport* (OT) distance, using projected gradient descent (projection via a hierarchical map). The learned ultrametric trees outperform several baselines on synthetic and real distributional data. Other optimization-driven approaches include differentiable losses on component trees (Perret & Cousty, 2022) (end-to-end learning of hierarchical segmentations), which directly tune the altitudes (merge levels) of a hierarchy.

Deep methods increasingly enforce ultrametric structure during training. Lapertot et al. (2024) introduce a differentiable ultrametric layer that maps predicted pairwise dissimilarities to an ultrametric, enabling end-to-end learning of hierarchical image segmentations with hierarchy-aware losses (e.g., hierarchical Rand index). In 3D vision (He et al., 2024), ultrametric feature fields impose an ultrametric contrastive loss so latent features satisfy the ultrametric inequality, yielding view-consistent hierarchical segmentations that outperform flat baselines. At a more foundational level, ultrametric neural networks (v-PuNNs) (N'guessan, 2025) with $p$-adic weights deliver transparent hierarchical representations: each neuron encodes a $p$-adic ball, guaranteeing a perfectly ultrametric output metric; empirically, these models recover large taxonomies with near-perfect leaf accuracy and zero triangle-inequality violations.

Collectively, these works treat the subdominant ultrametric not merely as a byproduct of single-linkage, but as a stable, computable projection from metrics to trees that supports temporal coherence, statistical stability of dendrograms, and fast algorithmic reductions in modern hierarchical clustering.

**Takeaway:** The above works show that the minmax (single-linkage) ultrametric is not a relic of classical clustering, but an actively used stability tool in modern hierarchical pipelines: it is chosen precisely because it behaves well under perturbations and admits clean algorithmic structure. Our results refine this picture along a complementary axis: instead of only controlling how *far* an ultrametric can move under metric noise, we control *how many* pairwise relations can change and which parts of the tree they can reach. In settings such as temporal HC or hierarchical graphical models that already rely on $u_d$ for robustness, our Hamming–Lipschitz bounds provide principled tools to (i) localize the effect of sparse perturbations and (ii) identify load-bearing cuts where instability or model misspecification is structurally concentrated.

### 1.3 Contributions:

At a high level, this paper makes the following contributions.

- **A sparsity-aware view of ultrametric stability.** We move beyond classical $\ell_\infty$ / Gromov–Hausdorff results and study the minmax ultrametric in a Hamming setting, where the cost of a perturbation is the *number* of pairwise distances that are edited, not how large the edits are.

- **Which pairs can actually change?** We show that sparse edits do *not* propagate arbitrarily through the hierarchy (Theorem 1). Instead, we give a simple structural rule that says exactly which pairs can be affected and when those pairs are guaranteed to remain unchanged.

- **A data-dependent Lipschitz constant.** For each edited pair, we define the sharp per-edit affected size, which localizes the pairs that can change through the cuts actually exposed by that edit (Theorem 2). We also introduce a tree-only envelope $\bar{L}_T$, obtained by maximizing the union of cut-rectangles along MST paths. This yields a structural bound on how many entries of $u_d$ can move under a sparse perturbation, while cleanly separating the edit-dependent sharp quantity from the tree-only global one.

- **Sharpness and worst-case behavior.** We show that our bound is attained exactly for tree-edge edits under strict cut separation and is attained on explicit off-tree families (Theorem 3). In particular, there are examples where changing a single pairwise distance forces a quadratic number of ultrametric entries to change, so no substantially smaller instance-independent bound is possible.

- **When changes almost add up?** For multi-edit perturbations, we give a conditional near-additivity principle: if one can certify per-edit changed regions that nearly saturate the exposed-region scores and have negligible aggregate overlap, then the total number of changed ultrametric entries is asymptotically close to the sum of those scores (Corollary 1).

- **Simple case studies as diagnostics.** Finally, we use the theorem-motivated structural score $S_{\text{union}}(e) = |A_e||B_e|$ as a diagnostic tool in two small case studies: *(i)* deep embeddings (DINO+UMAP) of CIFAR-10, ImageNet-10, and STL-10, and *(ii)* a superpixel segmentation of the Cameraman image. In both cases, high-score edges line up with empirically fragile parts of the hierarchy, illustrating that the theory can inform practical vulnerability maps even though the experiments are diagnostic rather than task-optimized.

*All proofs of our proposed theorems and corollaries are deferred to Appendix due to space constraints.*

## 2 Notation, Assumptions, and Preliminaries

Let $S = \{1, \ldots, n\}$ be a finite index set, and let $d : S \times S \to \mathbb{R}_{\geq 0}$ be a symmetric dissimilarity with $d(i, i) = 0$ for all $i \in S$. We write $\|\cdot\|$, $\|\cdot\|_\infty$, and $\|\cdot\|_0$ for the $\ell_2$ norm, the $\ell_\infty$ norm, and the $\ell_0$ pseudo-norm, respectively (always applied to the vector of upper-triangular entries unless stated otherwise). Because $\|\cdot\|_0$ counts upper-triangular matrix entries, all Hamming counts in this paper are over unordered pairs. We view $d$ in three equivalent ways: (i) as a function on ordered pairs $(i, j) \in S \times S$; (ii) as a symmetric matrix $\mathbf{D} \in \mathbb{R}^{n \times n}$ with entries $\mathbf{D}_{ij} = d(i, j)$; and (iii) as an edge-weight function on the complete undirected graph $G = (V, E)$ with vertex set $V = S$ and edge set $E = \{\{i, j\} : 1 \leq i < j \leq n\}$. When convenient, we write $d(e)$ for the weight of an edge $e = \{i, j\} \in E$. We define the set of all unordered pairs as $\binom{V}{2} := \{\{i, j\} : i, j \in V, \ i \neq j\}$. Let $T = (V, E(T))$ denote the unique MST of $d$ on $G$ (we assume a strict lexicographic tie-breaking rule to guarantee uniqueness; see Assumption 1 for details). For each tree edge $e = \{a, b\} \in E(T)$, we write $C_e = (A_e, B_e)$ for the fundamental cut obtained by removing $e$ from $T$, so that $A_e, B_e \subseteq V$ are the vertex sets of the two connected components of $T \setminus \{e\}$. We can then define the associated cut-pair set $\mathcal{R}_e := \{\{i, j\} \in \binom{V}{2} : i \in A_e, \ j \in B_e\}$. Thus $\mathcal{R}_e$ is the unordered-pair analogue of the rectangle $A_e \times B_e$.

We abbreviate $w_e := d(e)$ and define the alternative cut minimum

$$w_e^+(d) := \min\{d(x, y) : x \in A_e, \ y \in B_e, \ \{x, y\} \neq e\}, \tag{2}$$

and the corresponding cut gap

$$\Delta_e(d) := w_e^+(d) - w_e \geq 0. \tag{3}$$

To ensure a unique MST, we adopt the standard assumption of lexicographic tie-breaking: whenever two edges have identical weights, their order is resolved by a fixed, secondary lexicographic ordering (for instance,

based on endpoint indices or edge identifiers). This convention is not restrictive, it is a widely accepted device in both theoretical analyses and algorithmic implementations of MST-related problems. Recent works across theory and systems routinely employ the same assumption to guarantee determinism and analytical clarity, including sublinear and dynamic formulations (Patlin & van den Brand, 2025; de Vos & Grilnberger, 2025), massively parallel and distributed MST algorithms (Azarmehr et al., 2025; Sanders & Schimek, 2023), and polymatroid-based theoretical generalizations (Harb et al., 2023). Thus, lexicographic tie-breaking serves as a benign technical convention rather than a substantive limitation, ensuring well-definedness without affecting optimality or generality.

**Assumption 1.** *Fix a deterministic total order $\prec$ on edges (e.g., lexicographic). We compare edges by the lexicographic pair $(w(e), \mathrm{rank}_\prec(e))$. Equivalently, conceptually perturb*

$$w'(e) = w(e) + \eta \, \mathrm{rank}_\prec(e), \qquad \text{with a single infinitesimal } \eta > 0, \tag{4}$$

*and run Kruskal on $w'$ (ties in $w$ are broken by $\prec$). This yields a unique MST $T$ and a strictly ordered MST edge list. This infinitesimal perturbation is used only to select a unique MST and a deterministic processing order among equal-weight edges; all quantities $d(e)$, $u_d$, $w_e^+(d)$, and $\Delta_e(d)$ refer to the original unperturbed dissimilarity values unless explicitly stated otherwise.*

Under Assumption 1, the MST is unique and its tree edges are strictly ordered under the lexicographically perturbed weights. We next recall the minmax ultrametric and the standard MST facts used in our analysis. In general, $\Delta_e(d)$ need not be strictly positive: lexicographic tie-breaking guarantees a unique MST and a deterministic edge order, but it does not alter the original numeric edge weights. We say that a tree edge $e \in E(T)$ is strictly cut-separated if $\Delta_e(d) > 0$.

Given a perturbed dissimilarity $\tilde{d} : S \times S \to \mathbb{R}_{\geqslant 0}$, we measure the size of the perturbation by the number of edited pairs, i.e., in the $\ell_0$ sense:

$$\|d - \tilde{d}\|_0 := \#\big\{ \{i, j\} \subseteq S : i < j, \ d(i, j) \neq \tilde{d}(i, j) \big\}. \tag{5}$$

Let $(h_r)_{r=1}^{n-1}$ denote the increasing list of MST edge weights $h_r \in \{w_e : e \in E(T)\}$, and similarly $(\tilde{h}_r)_{r=1}^{n-1}$ for the MST of $\tilde{d}$.

**Ultrametric operator in index notation.** In the introduction we viewed the subdominant (min–max) ultrametric as the output of an operator

$$U : (X, d) \mapsto (X, u_d). \tag{6}$$

In this finite-index setting we work directly with the indexed version: for each dissimilarity $d$ on $S$, we denote by $u_d : S \times S \to \mathbb{R}_{\geqslant 0}$ the associated subdominant ultrametric, and we write

$$U(d) = u_d, \qquad u_d(i, j) \text{ for the } (i, j)\text{-entry of the ultrametric.} \tag{7}$$

All of our subsequent theorems and inequalities will be stated in terms of indices $i, j \in S$, the MST $T$ on $S$, and the entries of $d$ and $u_d$.

We now formally define the minmax ultrametric below:

**Definition 1** (Minmax subdominant ultrametric). *Given a finite set $S$ with dissimilarity function $d : S \times S \to \mathbb{R}_{\geqslant 0}$, the minmax subdominant ultrametric $u_d : S \times S \to \mathbb{R}_{\geqslant 0}$ is the largest ultrametric dominated by $d$. Explicitly, for any $i, j \in S$ with $i \neq j$, it is constructed by minimizing the bottleneck over all possible paths between $i$ and $j$:*

$$u_d(i, j) = \min_{P \in \mathcal{P}(i,j)} \max_{1 \leqslant t \leqslant m} d(v_{t-1}, v_t) \tag{8}$$

*where $\mathcal{P}(i, j)$ denotes the set of all finite paths $P = (v_0, v_1, \ldots, v_m)$ from $v_0 = i$ to $v_m = j$.*

### 2.1 Two MST Facts Used Throughout

Our analysis repeatedly uses two standard facts about the minimum spanning tree $T$ of $d$. These are common Lemmas on MST, one can easily find them in the standard algorithms textbook by Cormen et al. (2022).

**Lemma 1** (MST characterization). *If $T$ is any minimum spanning tree (MST) of the complete graph with weights given by the dissimilarity function $d$, then $u_d(i,j) = \max_{e \in \text{path}_T(i,j)} d(e) \quad \forall i \neq j$.*

**Lemma 2** (Cut property, with uniqueness). *Let $(A, B)$ be any cut of $V$ and let $e^\star \in E$ be an edge with one endpoint in $A$ and one in $B$. If $d(e^\star) < d(f)$ for every other cut edge $f$ across $(A, B)$, then $e^\star$ belongs to every MST of $d$.*

## 3 Sparsity-Localized Perturbations and Pairwise Impact

Our first theorem establishes a localization principle for the subdominant ultrametric under sparse edge edits. At the level of the MST, it shows that if the tree path between two points contains no edited tree edges, then their ultrametric value cannot increase, and it remains unchanged whenever no edited off-tree edge creates a cheaper crossing across any fundamental cut along that path. Globally, the theorem identifies a set of potentially affected tree edges and proves that every changed ultrametric entry must lie in the union of the associated cut-pair sets. This yields an explicit combinatorial upper bound on the support of $u_d - u_{\tilde{d}}$ in terms of the forest obtained by deleting those affected edges from the MST. For single-linkage clustering, the interpretation is that sparse perturbations can only propagate through edited or newly exposed cuts: subtrees separated from these cuts remain rigid, while only the corresponding branches of the hierarchy can move. Thus, Theorem 1 gives both a structural description of where changes may occur and a worst-case bound on how many pairwise merge heights can be altered by a sparse perturbation.

**Theorem 1** (Localization of ultrametric under sparse edge edits). *Let $d : \binom{V}{2} \mapsto \mathbb{R}_{\geqslant 0}$, and let $T = (V, E(T))$ be the (tie-broken under Assumption 1) MST of $d$. Let $F \subseteq \binom{V}{2}$ be a set of edited edges and let $\tilde{d}$ be any dissimilarity with $\tilde{d}(e) = d(e)$ for all $e \notin F$ (no restriction on $e \in F$).*

*For $i \neq j$ let $P_T(i,j)$ be the unique $i$–$j$ path in $T$. By Lemma 1, $u_d(i,j) = \max_{e \in P_T(i,j)} w_e$, and therefore the following holds:*

*(i) (Monotone upper bound, no edited $T$-edges on the path) If $P_T(i,j) \cap F = \varnothing$, then $u_{\tilde{d}}(i,j) \leq u_d(i,j)$.*

*(ii) (Sufficient conditions for equality) If $P_T(i,j) \cap F = \varnothing$ and for every $e \in P_T(i,j)$, all edited edges $f \in F$ crossing $C_e$ satisfy $\tilde{d}(f) \geqslant w_e$, then $u_{\tilde{d}}(i,j) = u_d(i,j)$.*

*(iii) (Pair-count bound for possible changes) Define the set of potentially affected MST edges*

$$\mathcal{E} := \left\{ e \in E(T) : e \in F \text{ or } \exists f \in F \text{ crossing } C_e \text{ with } \tilde{d}(f) < w_e \right\}. \tag{9}$$

*Then the number of unordered pairs whose ultrametric value changes satisfies*

$$\left| \left\{ \{i,j\} \in \binom{V}{2} : u_{\tilde{d}}(i,j) \neq u_d(i,j) \right\} \right| = \|u_d - u_{\tilde{d}}\|_0 \leq \left| \bigcup_{e \in \mathcal{E}} \mathcal{R}_e \right| = \binom{n}{2} - \sum_{t=1}^{m} \binom{|C_t|}{2}, \tag{10}$$

*where $C_1, \ldots, C_m$ are the vertex sets of the connected components of the forest $T - \mathcal{E}$.*

**Remark.** *Theorem 1 reveals an important asymmetry. If the tree path $P_T(i,j)$ contains no edited tree edge, then the original tree bottleneck remains available under $\tilde{d}$, so the ultrametric value $u_{\tilde{d}}(i,j)$ can never exceed $u_d(i,j)$. In that regime, a change can only occur through a strictly cheaper off-tree crossing of one of the fundamental cuts along $P_T(i,j)$. Thus sparse perturbations act in two qualitatively different ways: edits on the tree path can raise or lower merge heights directly, whereas off-tree edits can only lower them by opening cheaper alternative connections across MST cuts.*

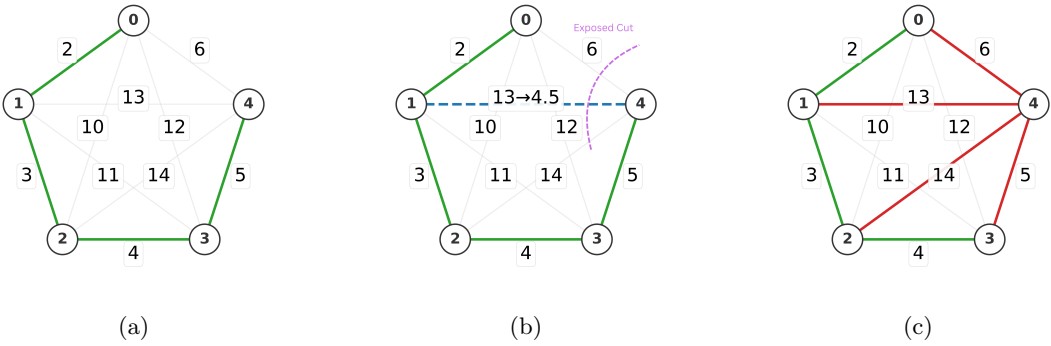

(a)            (b)            (c)

Figure 1: Illustration of Theorems 1 and 2. (a) Initial MST (green). (b) A sparse off-tree edit on edge $\{1, 4\}$ (dotted blue) that exposes the MST edge $\{4, 5\}$. (c) The affected region (red), given by the cut-pair set $\mathcal{R}_{\{4,5\}} = \big\{\{i, 4\} : i \in \{0, 1, 2, 3\}\big\}$, upper-bounds the total number of changed ultrametric entries, i.e. $\|u_d - u_{\tilde{d}}\|_0$.

Also, it gives a first coarse localization result: it shows that all changes in the ultrametric are confined to a union of cut-pair sets associated with a small set of MST edges determined by the edit pattern. However, this description is still global across edits and does not yet yield a Hamming–Lipschitz inequality that separates the contribution of individual edited pairs. To sharpen the picture, we pass to the level of a single edited pair $f = \{x, y\}$. For each such $f$, we isolate the subset of tree edges whose cuts are actually exposed by that edit; these are the exposed cuts $\Xi(f)$. Formally, for an edited pair $f = \{x, y\} \in F$, define the set of exposed cuts by $\Xi(f) := (\{f\} \cap E(T)) \cup \big\{e \in E(T) : f \text{ crosses } C_e \text{ and } \tilde{d}(f) < w_e\big\}$. The next theorem defines the sharp per-edit exposed-cut score

$$S_{\text{union}}(f) := \left| \bigcup_{e \in \Xi(f)} \mathcal{R}_e \right|, \tag{11}$$

and shows that every ultrametric entry changed by the edit must lie inside this exposed region. By further enlarging the exposed cuts to the full tree path $P_T(x, y)$, we obtain a tree-only global envelope $\bar{L}_T$, which yields a Hamming–Lipschitz type bound for arbitrary sparse perturbations.

---

**Theorem 2** (Hamming–Lipschitz bound via exposed cuts). *Let $d : \binom{V}{2} \to \mathbb{R}_{\geq 0}$ be a dissimilarity on a finite set $V$, and let $T = (V, E(T))$ be the (tie-broken under Assumption 1) MST of $d$. Let $\tilde{d}$ be any perturbed dissimilarity, and let $F := \big\{f \in \binom{V}{2} : \tilde{d}(f) \neq d(f)\big\}$ be its edit support. For an edited pair $f = \{x, y\} \in F$, define the set of exposed cuts by*

$$\Xi(f) := (\{f\} \cap E(T)) \cup \big\{e \in E(T) : f \text{ crosses } C_e \text{ and } \tilde{d}(f) < w_e\big\}. \tag{12}$$

*Let $C := \big\{\{i, j\} \in \binom{V}{2} : u_{\tilde{d}}(i, j) \neq u_d(i, j)\big\}$ denote the set of unordered pairs whose ultrametric values change. Further, define the sharp per-edit exposed-cut size $S_{\text{union}}(f) := \big|\bigcup_{e \in \Xi(f)} \mathcal{R}_e\big|$. For an edited pair $f = \{x, y\}$, define the tree-only path envelope $\bar{S}_T(f) := \big|\bigcup_{e \in P_T(x,y)} \mathcal{R}_e\big|$, and the global tree-only constant $\bar{L}_T := \max_{f \in \binom{V}{2}} \bar{S}_T(f)$.*
*Then:*

*(i) $C \subseteq \bigcup_{f \in F} \bigcup_{e \in \Xi(f)} \mathcal{R}_e$. Consequently, $\|u_d - u_{\tilde{d}}\|_0 = |C| \leq \big|\bigcup_{f \in F} \bigcup_{e \in \Xi(f)} \mathcal{R}_e\big|$.*

*(ii) for every edited pair $f \in F$,*
$$S_{\text{union}}(f) \leq \bar{S}_T(f) \leq \bar{L}_T, \tag{13}$$

*and therefore*

$$\|u_d - u_{\tilde{d}}\|_0 \leq \sum_{f \in F} S_{\text{union}}(f) \leq \sum_{f \in F} \bar{S}_T(f) \leq \bar{L}_T |F| = \bar{L}_T \|d - \tilde{d}\|_0. \tag{14}$$

*In particular,*

$$\bar{L}_T \leq \binom{|V|}{2}. \tag{15}$$

**Remark.** *Theorem 2 separates two distinct layers of control. The quantity $S_{\text{union}}(f)$ is the sharp, edit-dependent score: it records only those MST cuts that are actually exposed by the specific perturbation of $f$. In contrast, $\bar{S}_T(f)$ and $\bar{L}_T$ are tree-only envelopes obtained by enlarging the exposed region to the full MST path of the edited pair. Thus the theorem distinguishes the mechanism of propagation, which is localized through exposed cuts, from a coarser but globally uniform capacity of the tree to transmit sparse perturbations.*

It yields a Hamming–Lipschitz upper bound that is both localized and structurally interpretable: sparse perturbations can propagate only through exposed MST cuts, and their total effect is controlled by the associated union of cut-pair sets. This naturally raises the next question: are these quantities merely proof-level upper bounds, or do they capture the true scale of instability of the subdominant ultrametric? In particular, one would like to know whether the sharp per-edit score $S_{\text{union}}(f)$ can actually be attained, and whether the resulting dependence on tree geometry is intrinsic. Theorem 3 answers this by showing exact attainability for tree-edge edits under strict cut separation, and sharpness on explicit off-tree families, including worst-case examples in which a single sparse edit changes $\Theta(n^2)$ ultrametric entries.

**Theorem 3** (Analysis of the Hamming–Lipschitz bound). *Let $d : \binom{V}{2} \to \mathbb{R}_{\geq 0}$ be a dissimilarity and let $T = (V, E(T))$ be the (tie-broken under Assumption 1) MST of $d$. For an edited pair $f = \{x, y\}$ and edited dissimilarity $\tilde{d}$ with $\tilde{d}(g) = d(g)$ for $g \notin \{f\}$, define*

$$\Xi(f) := (\{f\} \cap E(T)) \cup \{e \in E(T) : f \text{ crosses } C_e \text{ and } \tilde{d}(f) < w_e\}, \qquad S_{\text{union}}(f) := \left| \bigcup_{e \in \Xi(f)} \mathcal{R}_e \right|. \tag{16}$$

*Then:*

*(i) (Tree-edge edit under strict cut separation) If $f = e \in E(T)$ and $e$ is strictly cut-separated, i.e. $\Delta_e(d) = w_e^+(d) - w_e > 0$, then there exists $\tilde{d}$ supported on $\{f\}$ such that*

$$\|u_d - u_{\tilde{d}}\|_0 = |A_e||B_e| = S_{\text{union}}(f). \tag{17}$$

*(ii) (Off-tree sharpness on an explicit family) There exist dissimilarities $d$, off-tree edges $f \notin E(T)$, and single-edge edits $\tilde{d}$ supported on $\{f\}$ such that*

$$\|u_d - u_{\tilde{d}}\|_0 = \left| \bigcup_{e \in \Xi(f)} \mathcal{R}_e \right| = S_{\text{union}}(f) = \Theta(n^2). \tag{18}$$

*Thus, the upper bound of Theorem 2 is attained on explicit off-tree instances, and in the worst case a single off-tree edit can force a quadratic number of ultrametric changes.*

*(iii) (Necessity of instance dependence) Consequently, no universal subquadratic function $c(n) = o(n^2)$ can satisfy*

$$\|u_d - u_{\tilde{d}}\|_0 \leq c(n) \|d - \tilde{d}\|_0 \tag{19}$$

*for all instances.*

**Remark.** *Theorem 3 shows that the exposed-cut quantities from Theorem 2 are not artifacts of the proof. In particular, the instability of the subdominant ultrametric is governed by the geometry of the MST itself: a single sparse edit can trigger changes on the scale of an entire cut-pair set, and in explicit off-tree families this propagation reaches quadratic size. Thus the dependence on tree geometry is intrinsic to the operator, rather than a byproduct of our bounding technique.*

Theorem 3 resolves the single-edit case: it shows that the exposed-cut score can be attained exactly in structured settings and that, in the worst case, even one edited distance can induce $\Theta(n^2)$ ultrametric changes. *The next natural question is how multiple sparse edits interact.* If their exposed regions substantially overlap, the total number of changed entries may be far smaller than the sum of the individual scores; if they are largely disjoint, one expects an approximately additive effect. Corollary 1 formalizes this latter regime in a conditional form: whenever one can certify large per-edit changed regions with negligible aggregate overlap, the overall Hamming change is asymptotically close to the sum of the corresponding exposed-cut scores.

---

**Corollary 1.** *Fix a minimum spanning tree $T$ of $d$, an edit set $F \subseteq \binom{V}{2}$, and edited weights $\tilde{d}$ supported on $F$. For each $f \in F$, define the exposed region $R(f) := \bigcup_{e \in \Xi(f)} \mathcal{R}_e$, so that $|R(f)| = S_{\text{union}}(f)$. Assume that for each $f \in F$ there exists a certified changed-pair set $Q(f) \subseteq R(f)$ such that every pair in $Q(f)$ indeed changes under the common edited dissimilarity $\tilde{d}$, i.e. $Q(f) \subseteq \{\{i,j\} \in \binom{V}{2} : u_{\tilde{d}}(i,j) \neq u_d(i,j)\}$. Then:*

(i)
$$\left| \bigcup_{f \in F} Q(f) \right| \leq \|u_d - u_{\tilde{d}}\|_0 \leq \left| \bigcup_{f \in F} R(f) \right| \leq \sum_{f \in F} S_{\text{union}}(f).$$

(ii) *Moreover, consider any asymptotic regime of instances (for example, $|V| = n \to \infty$) in which*
$$\sum_{f \in F} |Q(f)| = (1 - o(1)) \sum_{f \in F} S_{\text{union}}(f),$$

*and the certified regions have asymptotically negligible total overlap:*
$$\sum_{\substack{f,f' \in F \\ f < f'}} |Q(f) \cap Q(f')| = o\left( \sum_{f \in F} |Q(f)| \right).$$

*Then*
$$\|u_d - u_{\tilde{d}}\|_0 = (1 - o(1)) \sum_{f \in F} S_{\text{union}}(f).$$

---

**Remark.** *Corollary 1 isolates two logically distinct tasks. The first is a* geometric *task: identify exposed regions $R(f)$ that contain all pairs that could possibly change. The second is a* certification *task: exhibit subsets $Q(f) \subseteq R(f)$ whose pairs are guaranteed to change under the common perturbation. Once such certified regions are available, the global Hamming count is governed purely by set overlap. In this sense, the corollary turns the multi-edit problem from one of ultrametric analysis into one of certifiable combinatorial packing.*

Corollary 1 completes the picture by showing how the single-edit geometry from Theorems 1–3 extends to sparse multi-edit regimes. The key issue is no longer the effect of an individual edit, but the interaction among their exposed regions. When these regions overlap heavily, different edits compete for the same ultrametric entries and the total effect can be far smaller than the sum of their individual scores. When one can instead certify large per-edit changed regions with negligible aggregate overlap, the total Hamming change becomes asymptotically additive. Thus the corollary identifies the precise structural mechanism behind near-additivity: it is not sparsity alone, but sparsity together with low-overlap exposure in the MST geometry.

## 4 Empirical Case Study: Vulnerability Maps of Deep Embeddings

Our contribution is primarily theoretical, so the role of this section is diagnostic rather than benchmark-driven. Theorems 1 and 2 show that sparse metric edits can change the ultrametric only through edited or newly exposed MST cuts, and that the extent of propagation is controlled by the associated cut-pair sets. This suggests a concrete empirical question: *in representation graphs arising from real data, are there only a few load-bearing MST edges whose perturbation causes substantial ultrametric damage, while most edges are comparatively harmless?* The experiment below is designed to probe exactly that question.

**Motivation from the theory.** For a tree edge $e \in E(T)$, Theorem 2 reduces the per-edit exposed region to the single cut-rectangle associated with $e$, whose cardinality is $S_{\text{union}}(e) = |A_e| |B_e|$. Thus $S_{\text{union}}(e)$ is the natural structural score predicted by the theory: it quantifies how many unordered pairs can potentially be affected when the hierarchy is stressed across that cut. Our main empirical goal is therefore to test whether this score meaningfully ranks edges by realized vulnerability in real representation graphs.

### 4.1 Protocol

We use three 10-class image datasets: CIFAR-10, ImageNet-10, and STL-10. For each dataset, we start from precomputed DINO-ViT features, apply UMAP to obtain a 20-dimensional embedding, and subsample a few thousand points for efficiency. On each embedding, we build the complete Euclidean graph, compute its tie-broken MST $T$, and obtain the induced subdominant ultrametric $u_d$ via the MST bottleneck formula. We then compute the structural score $S_{\text{union}}(e) = |A_e||B_e|$ for every tree edge $e \in E(T)$. Theorem 1 predicts that ultrametric changes must remain localized to unions of such cut-rectangles, while Theorem 2 identifies $S_{\text{union}}(e)$ as the sharp tree-edge quantity controlling the size of the exposed region. Figure 2 therefore serves two purposes: the bottom row visualizes the empirical distribution of this theorem-motivated score across the MST, and the top row evaluates whether targeting high-score edges indeed produces larger realized ultrametric damage.

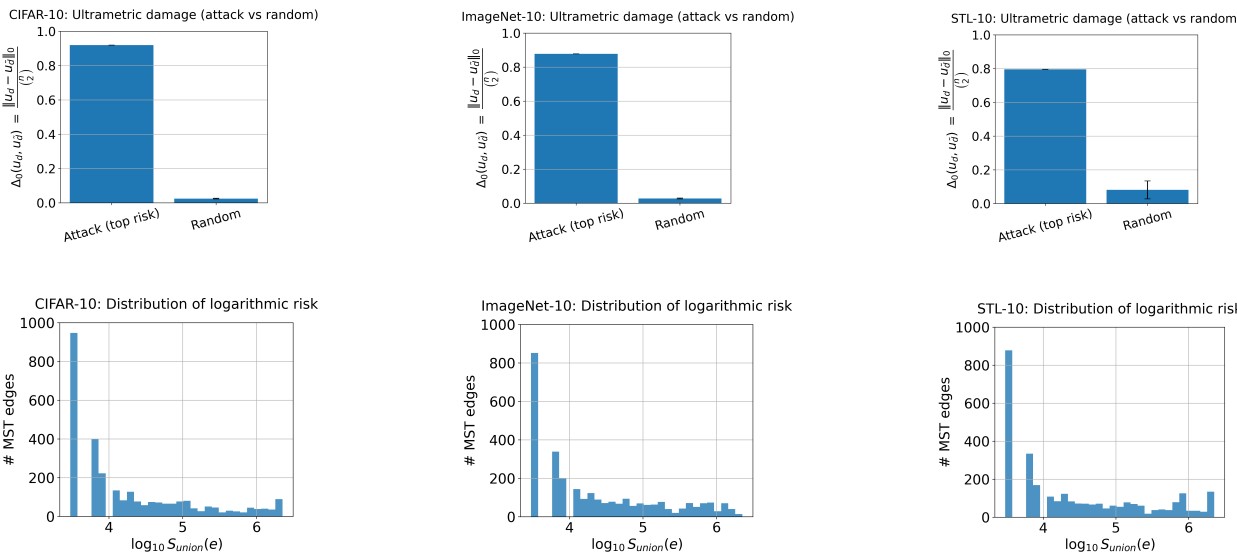

Figure 2: **Top row:** normalized ultrametric damage $\Delta_0(u_d, u_{\tilde{d}})$ under targeted (top-risk) versus random sparse perturbations for CIFAR-10, ImageNet-10, and STL-10. **Bottom row:** histograms of the theorem-motivated structural score $\log_{10} S_{\text{union}}(e)$ over MST edges for the same three datasets. It shows that structural risk is concentrated in a small high-score tail, while the top row shows that targeting that tail causes substantially larger realized Hamming damage than random edits with the same budget, consistent with Theorems 1 and 2.

To test this, we fix a sparse edit budget $m$ equal to 10% of the MST edges and compare two strategies:

- **Structural attack (top-risk):** choose the $m$ tree edges with largest $S_{\text{union}}(e)$.

- **Random baseline:** choose $m$ tree edges uniformly at random from the same pool.

In both cases, we construct a perturbed metric $\tilde{d}$ by sharply inflating the selected tree-edge weights, recompute the MST and the induced ultrametric $u_{\tilde{d}}$, and measure the normalized Hamming distortion

$$\Delta_0(u_d, u_{\tilde{d}}) := \frac{\|u_d - u_{\tilde{d}}\|_0}{\binom{n}{2}}. \tag{20}$$

This is a stress test inspired by the tree-edge exposed-cut analysis. It is not intended as a literal validation of the sharpness theorem edge-by-edge, but rather as an empirical probe of whether the score from Theorem 2 tracks practical vulnerability.

### 4.2 Results

Figure 2 (bottom row) shows that the distribution of $\log_{10} S_{\text{union}}(e)$ is strongly skewed for all three datasets. Most MST edges lie in a low-score bulk, corresponding to leaf-like or weakly load-bearing cuts, while a small number of bridge-like edges form a thin high-score tail. This is precisely the type of heterogeneity suggested by Theorems 1 and 2: sparse perturbations should not propagate uniformly through the hierarchy, but instead concentrate around a small set of structurally important cuts.

Figure 2 (top row) shows that this structural heterogeneity is not merely combinatorial. Across CIFAR-10, ImageNet-10, and STL-10, the top-risk attack consistently induces much larger normalized Hamming damage than the random baseline at the same edit budget. In particular, even in budgets where random edits leave $\Delta_0(u_d, u_{\tilde{d}})$ close to zero, editing the highest-score edges already flips a visible fraction of ultrametric entries. The practical conclusion is that the theorem-motivated score $S_{\text{union}}(e)$ identifies a small set of load-bearing edges whose perturbation produces disproportionate global impact.

Overall, this case study should be read as a diagnostic interpretation of the theory. Theorem 1 explains *where* propagation can occur, while Theorem 2 supplies the natural per-edge structural score. Figure 2 shows that, on real deep-embedding graphs, these quantities yield a meaningful vulnerability map: most MST edges are structurally benign, but a small high-score tail captures the edges whose sparse perturbation can ripple widely through the induced hierarchy. Taken together, these experiments provide an empirical counterpart to the theoretical picture: the structural score $S_{\text{union}}(e)$ identifies a small set of load-bearing MST edges whose perturbation has disproportionate global impact, while most edges are comparatively benign.

*Beyond the deep-embedding vulnerability maps presented above, we further evaluate the practical utility of the structural score in two downstream settings. Appendix C presents an MST-based superpixel segmentation case study, showing that restricting sparse edits to low-$S_{\text{union}}(e)$ edges better preserves the induced clustering than local-weight heuristics. In Appendix D, we also include a budgeted active MST-edge verification task, specifically to provide an ML-relevant use case of our theory. There, $S_{\text{union}}(e)$ is used to prioritize which tree edges should be checked under limited supervision, testing whether the theory-derived score yields a cheap and effective query rule on a fixed sparse single-linkage backbone.*

## 5 Conclusion

We developed a sparsity-aware stability theory for the subdominant (minmax) ultrametric, complementing classical $\ell_\infty$/Gromov–Hausdorff results with an $\ell_0$-type perspective that controls the extent of change under sparse edits. Our analysis shows that propagation is mediated by the MST: only pairs whose tree paths traverse edited or newly exposed cuts can change, yielding a localization in terms of exposed cut-pair sets, a sharp per-edit score $S_{\text{union}}(f)$, and a tree-only global envelope $\bar{L}_T$. We further proved that this instance dependence is unavoidable by exhibiting single-edit constructions with $\Theta(n^2)$ changes, and identified explicit single-edit sharpness examples together with conditional multi-edit regimes in which the upper bound is

asymptotically tight. Together, these results offer a structural, interpretable account of how local perturbations can (or cannot) ripple through single-linkage hierarchies, and motivate using the associated risk scores as practical diagnostics for locating robust versus load-bearing edges in real data.

## Acknowledgement

Alokendu Mazumder is supported through the Kotak-IISc AI-ML Centre at IISc Bengaluru.

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

# Supplementary Material

## Contents

This *supplementary material* is part of the submission *On Hamming–Lipschitz Stability of the Subdominant (Minmax) Ultrametric: Theory and Simple Proofs*

# A    Proofs of Technical Lemmas

## A.1    Proof of Lemma 1

**Lemma 3** (MST characterization)**.** *If $T$ is any minimum spanning tree (MST) of the complete graph with weights given by the dissimilarity function $d$, then $u_d(i,j) = \max_{e \in \mathrm{path}_T(i,j)} d(e) \quad \forall i \neq j$.*

*Proof.* Fix $i \neq j$ and let $P_T(i,j)$ be the unique $i$–$j$ path in $T$. Let

$$\lambda := \max_{e \in P_T(i,j)} d(e),$$

and choose an edge $e^\star \in P_T(i,j)$ with $d(e^\star) = \lambda$. Removing $e^\star$ splits $T$ into two components $A$ and $B$ with $i \in A$ and $j \in B$.

Because $T$ is an MST, the tree edge $e^\star$ is a minimum-weight edge across its fundamental cut $(A,B)$. Hence every path $P$ from $i$ to $j$ must contain some edge crossing $(A,B)$, and every such crossing edge has weight at least $\lambda$. Therefore

$$\max_{e \in P} d(e) \geq \lambda \qquad \text{for every path } P \in \mathcal{P}(i,j),$$

which implies

$$u_d(i,j) = \min_{P \in \mathcal{P}(i,j)} \max_{e \in P} d(e) \geq \lambda.$$

On the other hand, the specific tree path $P_T(i,j)$ has bottleneck exactly $\lambda$, so

$$u_d(i,j) \leq \max_{e \in P_T(i,j)} d(e) = \lambda.$$

Combining the two inequalities yields

$$u_d(i,j) = \max_{e \in P_T(i,j)} d(e),$$

as claimed. $\qquad\square$

## A.2    Proof of Lemma 2

**Lemma 4** (Cut property, with uniqueness)**.** *Let $(A,B)$ be any cut of $V$ and let $e^\star \in E$ be an edge with one endpoint in $A$ and one in $B$. If $d(e^\star) < d(f)$ for every other cut edge $f$ across $(A,B)$, then $e^\star$ belongs to every MST of $d$.*

*Proof.* Suppose, toward a contradiction, that there exists an MST $T' = (V, E(T'))$ with $e^\star \notin E(T')$. Then $T' \cup \{e^\star\}$ contains a unique simple cycle $C$. Since $e^\star$ crosses $(A,B)$, any cycle crosses a cut an even number of times; hence there exists an edge $f \in C \cap E$ with $f \neq e^\star$ that also crosses $(A,B)$. By the strict minimality of $e^\star$ across the cut, $d(f) > d(e^\star)$. Define

$$E'' := E(T') \cup \{e^\star\} \setminus \{f\}. \tag{21}$$

Then $E''$ is acyclic and has $|V| - 1$ edges, so $(V, E'')$ is a spanning tree. Its total weight satisfies

$$\sum_{e \in E''} d(e) = \sum_{e \in E(T')} d(e) + d(e^\star) - d(f) < \sum_{e \in E(T')} d(e), \tag{22}$$

contradicting the minimality of $T'$. Hence every MST must contain $e^\star$. $\qquad\square$

# B    Proofs of Major Theorems and Corollary

## B.1    Proof of Theorem 1

---

**Theorem 1** (Localization of ultrametric under sparse edge edits). *Let* $d : \binom{V}{2} \mapsto \mathbb{R}_{\geq 0}$, *and let* $T = (V, E(T))$ *be the (tie-broken under Assumption 1) MST of d. For* $e = \{a, b\} \in E(T)$ *let* $C_e = (A_e, B_e)$ *be its fundamental cut in T, and write* $w_e := d(e)$. *we define the associated cut-pair set* $\mathcal{R}_e := \big\{\{i, j\} \in \binom{V}{2} : i \in A_e,\ j \in B_e\big\}$. *Let* $F \subseteq \binom{V}{2}$ *be a set of edited edges and let* $\tilde{d}$ *be any dissimilarity with* $\tilde{d}(e) = d(e)$ *for all* $e \notin F$ *(no restriction on* $e \in F$).

*For* $i \neq j$ *let* $P_T(i, j)$ *be the unique* $i$–$j$ *path in T. From MST bottleneck representation* $u_d(i, j) = \max_{e \in P_T(i,j)} w_e$, *the following holds:*

(i) *(Monotone upper bound, no edited T-edges on the path) If* $P_T(i, j) \cap F = \varnothing$, *then* $u_{\tilde{d}}(i, j) \leq u_d(i, j)$.

(ii) *(Sufficient conditions for equality) If* $P_T(i, j) \cap F = \varnothing$ *and for every* $e \in P_T(i, j)$, *all edited edges* $f \in F$ *crossing* $C_e$ *satisfy* $\tilde{d}(f) \geq w_e$, *then* $u_{\tilde{d}}(i, j) = u_d(i, j)$.

(iii) *(Pair-count bound for possible changes) Define the set of potentially affected MST edges*

$$\mathcal{E} := \big\{e \in E(T) : e \in F \text{ or } \exists f \in F \text{ crossing } C_e \text{ with } \tilde{d}(f) < w_e\big\}. \tag{23}$$

*Then the number of unordered pairs whose ultrametric value changes satisfies*

$$\Big|\big\{\{i, j\} \in \binom{V}{2} : u_{\tilde{d}}(i, j) \neq u_d(i, j)\big\}\Big| = \|u_d - u_{\tilde{d}}\|_0 \leq \Big|\bigcup_{e \in \mathcal{E}} \mathcal{R}_e\Big| = \binom{n}{2} - \sum_{t=1}^{m} \binom{|C_t|}{2}, \tag{24}$$

*where* $C_1, \ldots, C_m$ *are the vertex sets of the connected components of the forest* $T - \mathcal{E}$.

---

*Proof.* (i) If $P_T(i, j) \cap F = \varnothing$, then all edges on the $T$-path are unchanged, i.e. $\tilde{d}(e) = d(e)$ for all $e \in P_T(i, j)$. Hence,

$$\begin{aligned} u_{\tilde{d}}(i, j) &= \min_{P \in \mathcal{P}(i,j)} \max_{f \in P} \tilde{d}(f) \\ &\leq \max_{e \in P_T(i,j)} \tilde{d}(e) = \max_{e \in P_T(i,j)} d(e) = u_d(i, j). \end{aligned} \tag{25}$$

(ii) Fix distinct points $i, j$ such that the MST path $P_T(i, j)$ contains no edited edges. Let $e \in P_T(i, j)$ be arbitrary, and let $P$ be any $i$–$j$ path in the complete graph. Since deleting $e$ separates $i$ and $j$, the path $P$ must cross the fundamental cut $C_e$. Let $g$ be any edge of $P$ that crosses $C_e$.

We claim that

$$\tilde{d}(g) \geq w_e. \tag{26}$$

Indeed, there are two cases.

If $g \in F$, then $g$ is an edited edge crossing $C_e$. By assumption, every edited edge crossing $C_e$ has weight at least $w_e$ under $\tilde{d}$. In particular,

$$\tilde{d}(g) \geq w_e. \tag{27}$$

If $g \notin F$, then $\tilde{d}(g) = d(g)$. Moreover, since $e$ is an MST edge defining the fundamental cut $C_e$, every edge crossing $C_e$ has weight at least $w_e$; otherwise replacing $e$ by a strictly lighter crossing edge would produce a lighter spanning tree (cut-property argument; cf. Lemma 2). Hence

$$\tilde{d}(g) = d(g) \geq w_e. \tag{28}$$

Thus in all cases the crossing edge $g$ on $P$ satisfies $\tilde{d}(g) \geq w_e$, and therefore

$$\max_{f \in P} \tilde{d}(f) \geq \tilde{d}(g) \geq w_e \qquad \text{for all } e \in P_T(i, j). \tag{29}$$

Consequently,

$$\max_{f \in P} \tilde{d}(f) \;\geqslant\; \max_{e \in P_T(i,j)} w_e \;=\; u_d(i,j), \tag{30}$$

and therefore

$$u_{\tilde{d}}(i,j) = \min_{P \in \mathcal{P}(i,j)} \max_{f \in P} \tilde{d}(f) \;\geqslant\; u_d(i,j). \tag{31}$$

Together with part (i), which gives $u_{\tilde{d}}(i,j) \leqslant u_d(i,j)$, we conclude that

$$u_{\tilde{d}}(i,j) = u_d(i,j). \tag{32}$$

*(iii)* We will prove the following statements $(P, Q, R)$ to complete the proof:

- $P$ : $\left\{ \{i,j\} \in \binom{V}{2} : u_{\tilde{d}}(i,j) \neq u_d(i,j) \right\} \subseteq \bigcup_{e \in \mathcal{E}} \mathcal{R}_e$.

- $Q$ : $\bigcup_{e \in \mathcal{E}} \mathcal{R}_e = \left\{ \{i,j\} \in \binom{V}{2} : i \text{ and } j \text{ in different components of } T - \mathcal{E} \right\}$.

- $R$ : $\left| \bigcup_{e \in \mathcal{E}} \mathcal{R}_e \right| = \binom{n}{2} - \sum_{t=1}^{m} \binom{|C_t|}{2}$.

Proof of (P): Fix $\{i,j\} \in \binom{V}{2}$ such that

$$u_{\tilde{d}}(i,j) \neq u_d(i,j). \tag{33}$$

We show that $\{i,j\} \in \mathcal{R}_e$ for some $e \in \mathcal{E}$.

First suppose that

$$P_T(i,j) \cap F \neq \varnothing. \tag{34}$$

Choose any edge $e \in P_T(i,j) \cap F$. Since $e$ is an edited tree edge, by definition

$$e \in \mathcal{E}. \tag{35}$$

Because $e$ lies on the unique tree path from $i$ to $j$, deleting $e$ separates $i$ and $j$, and hence

$$\{i,j\} \in \mathcal{R}_e. \tag{36}$$

Now suppose instead that

$$P_T(i,j) \cap F = \varnothing. \tag{37}$$

Since

$$u_{\tilde{d}}(i,j) \neq u_d(i,j), \tag{38}$$

the conclusion of part (ii) fails. Part (ii) states that if

$$P_T(i,j) \cap F = \varnothing \tag{39}$$

and if for every $e \in P_T(i,j)$ every edited edge crossing $C_e$ satisfies

$$\tilde{d}(f) \geqslant w_e, \tag{40}$$

then

$$u_{\tilde{d}}(i,j) = u_d(i,j). \tag{41}$$

Therefore, since the first condition already holds in the present case, the second condition must fail. Hence there exist some edge $e \in P_T(i,j)$ and some edited edge $f \in F$ crossing $C_e$ such that

$$\tilde{d}(f) < w_e. \tag{42}$$

Therefore, by the definition of $\mathcal{E}$,

$$e \in \mathcal{E}. \tag{43}$$

Again, because $e$ lies on the unique tree path from $i$ to $j$, deleting $e$ separates $i$ and $j$, so

$$\{i, j\} \in \mathcal{R}_e. \tag{44}$$

In either case, $\{i, j\}$ belongs to $\mathcal{R}_e$ for some $e \in \mathcal{E}$. Therefore

$$\left\{\{i, j\} \in \binom{V}{2} : u_{\tilde{d}}(i, j) \neq u_d(i, j)\right\} \subseteq \bigcup_{e \in \mathcal{E}} \mathcal{R}_e. \tag{45}$$

Proof of (Q): If $\{i, j\} \in \mathcal{R}_e$ for some $e \in \mathcal{E}$, then removing $e$ separates $i$ and $j$ in $T$, so they lie in different connected components of $T - \mathcal{E}$. Conversely, if $i$ and $j$ lie in different components of $T - \mathcal{E}$, then the unique tree path $P_T(i, j)$ contains at least one edge $e \in \mathcal{E}$. For that edge, one endpoint lies in $A_e$ and the other in $B_e$, so

$$\{i, j\} \in \mathcal{R}_e. \tag{46}$$

Hence

$$\bigcup_{e \in \mathcal{E}} \mathcal{R}_e = \left\{\{i, j\} \in \binom{V}{2} : i \text{ and } j \text{ lie in different connected components of } T - \mathcal{E}\right\}. \tag{47}$$

Proof of (R): There are $\binom{n}{2}$ unordered pairs in total. The unordered pairs not in $\bigcup_{e \in \mathcal{E}} \mathcal{R}_e$ are exactly those whose two endpoints lie in the same component $C_t$ of $T - \mathcal{E}$, and there are $\binom{|C_t|}{2}$ such pairs inside $C_t$. Summing over components and subtracting yields

$$\left|\bigcup_{e \in \mathcal{E}} \mathcal{R}_e\right| = \binom{n}{2} - \sum_{t=1}^{m} \binom{|C_t|}{2}. \tag{48}$$

Combining the previous steps, we obtain

$$\left|\left\{\{i, j\} \in \binom{V}{2} : u_{\tilde{d}}(i, j) \neq u_d(i, j)\right\}\right| = \|u_d - u_{\tilde{d}}\|_0 \leq \left|\bigcup_{e \in \mathcal{E}} \mathcal{R}_e\right| = \binom{n}{2} - \sum_{t=1}^{m} \binom{|C_t|}{2}. \tag{49}$$

$\square$

## B.2 Proof of Theorem 2

**Theorem 2** (Hamming–Lipschitz bound via exposed cuts). *Let $d : \binom{V}{2} \to \mathbb{R}_{\geq 0}$ be a dissimilarity on a finite set $V$, and let $T = (V, E(T))$ be the (tie-broken under Assumption 1) MST of $d$. For each tree edge $e = \{a, b\} \in E(T)$, let $C_e = (A_e, B_e)$ be its fundamental cut in $T$, and write $w_e := d(e)$. Let $\tilde{d}$ be any perturbed dissimilarity, and let $F := \{f \in \binom{V}{2} : \tilde{d}(f) \neq d(f)\}$ be its edit support. For an edited pair $f = \{x, y\} \in F$, define the set of exposed cuts by*

$$\Xi(f) := (\{f\} \cap E(T)) \cup \{e \in E(T) : f \text{ crosses } C_e \text{ and } \tilde{d}(f) < w_e\}. \tag{50}$$

*For each tree edge $e \in E(T)$, the set of associated cut-pairs $\mathcal{R}_e := \{\{i, j\} \in \binom{V}{2} : i \in A_e, \ j \in B_e\}$. Let $C := \{\{i, j\} \in \binom{V}{2} : u_{\tilde{d}}(i, j) \neq u_d(i, j)\}$ denote the set of unordered pairs whose ultrametric values change. Further, define the sharp per-edit exposed-cut size $S_{\text{union}}(f) := \left|\bigcup_{e \in \Xi(f)} \mathcal{R}_e\right|$. For an edited pair $f = \{x, y\}$, define the tree-only path envelope $\bar{S}_T(f) := \left|\bigcup_{e \in P_T(x, y)} \mathcal{R}_e\right|$, and the global tree-only constant $\bar{L}_T := \max_{f \in \binom{V}{2}} \bar{S}_T(f)$.*
*Then:*

*(i) $C \subseteq \bigcup_{f \in F} \bigcup_{e \in \Xi(f)} \mathcal{R}_e$. Consequently, $\|u_d - u_{\tilde{d}}\|_0 = |C| \leq \left|\bigcup_{f \in F} \bigcup_{e \in \Xi(f)} \mathcal{R}_e\right|$.*

*(ii) for every edited pair $f \in F$,*

$$S_{\text{union}}(f) \leq \bar{S}_T(f) \leq \bar{L}_T, \tag{51}$$

*and therefore*

$$\|u_d - u_{\tilde{d}}\|_0 \leq \sum_{f \in F} S_{\text{union}}(f) \leq \sum_{f \in F} \bar{S}_T(f) \leq \bar{L}_T |F| = \bar{L}_T \|d - \tilde{d}\|_0. \tag{52}$$

*In particular,*

$$\bar{L}_T \leq \binom{|V|}{2}. \tag{53}$$

*Proof. (i)* We first prove the localization statement. Let

$$\widehat{E} := \bigcup_{f \in F} \Xi(f) \subseteq E(T). \tag{54}$$

By the definition of $\Xi(f)$, a tree edge $e \in E(T)$ belongs to $\widehat{E}$ if and only if at least one of the following holds:

$$e \in F \cap E(T), \tag{55}$$

or

$$\exists f \in F \text{ such that } f \text{ crosses } C_e \text{ and } \tilde{d}(f) < w_e. \tag{56}$$

Hence

$$\widehat{E} = \left\{ e \in E(T) : e \in F \cap E(T) \text{ or } \exists f \in F \text{ crossing } C_e \text{ with } \tilde{d}(f) < w_e \right\}. \tag{57}$$

But this is exactly the set of potentially affected tree edges appearing in Theorem 1-(iii). Therefore, by Theorem 1-(iii), the changed-pair set satisfies

$$C \subseteq \bigcup_{e \in \widehat{E}} \mathcal{R}_e. \tag{58}$$

Since $\widehat{E} = \bigcup_{f \in F} \Xi(f)$, we obtain

$$C \subseteq \bigcup_{f \in F} \bigcup_{e \in \Xi(f)} \mathcal{R}_e, \tag{59}$$

which proves the claimed localization. Taking cardinalities yields

$$\|u_d - u_{\tilde{d}}\|_0 = |C| \leq \left| \bigcup_{f \in F} \bigcup_{e \in \Xi(f)} \mathcal{R}_e \right|. \tag{60}$$

*(ii)* We now derive the per-edit bound. By definition,

$$S_{\text{union}}(f) = \left| \bigcup_{e \in \Xi(f)} \mathcal{R}_e \right|. \tag{61}$$

Applying the union bound to the previous inclusion gives

$$\|u_d - u_{\tilde{d}}\|_0 \leq \sum_{f \in F} \left| \bigcup_{e \in \Xi(f)} \mathcal{R}_e \right| = \sum_{f \in F} S_{\text{union}}(f). \tag{62}$$

It remains to compare the sharp exposed-cut score with a tree-only envelope. Fix an edited pair $f = \{x, y\} \in F$. We claim that every exposed cut for $f$ lies on the tree path $P_T(x, y)$:

$$\Xi(f) \subseteq P_T(x, y). \tag{63}$$

Indeed, if $e \in \Xi(f) \cap E(T)$ because $e = f$, then trivially $e$ lies on $P_T(x, y)$. Otherwise, $e \in \Xi(f)$ means that $f = \{x, y\}$ crosses the cut $C_e = (A_e, B_e)$, so one endpoint of $f$ lies in $A_e$ and the other lies in $B_e$. In a tree, removing $e$ separates $x$ and $y$ if and only if $e$ lies on the unique path between them. Therefore

$$e \in P_T(x, y). \tag{64}$$

This proves the claim.

Hence

$$\bigcup_{e \in \Xi(f)} \mathcal{R}_e \subseteq \bigcup_{e \in P_T(x, y)} \mathcal{R}_e. \tag{65}$$

Taking cardinalities gives

$$S_{\text{union}}(f) \leq \bar{S}_T(f). \tag{66}$$

By the definition of $\bar{L}_T$,

$$\bar{S}_T(f) \leq \bar{L}_T \tag{67}$$

for every $f \in \binom{V}{2}$. Therefore

$$S_{\text{union}}(f) \leq \bar{S}_T(f) \leq \bar{L}_T. \tag{68}$$

Summing over $f \in F$ yields

$$\|u_d - u_{\tilde{d}}\|_0 \leq \sum_{f \in F} S_{\text{union}}(f) \leq \sum_{f \in F} \bar{S}_T(f) \leq \bar{L}_T \, |F|. \tag{69}$$

Since $F$ is exactly the support of the perturbation,

$$|F| = \|d - \tilde{d}\|_0, \tag{70}$$

and therefore

$$\|u_d - u_{\tilde{d}}\|_0 \leq \bar{L}_T \, \|d - \tilde{d}\|_0. \tag{71}$$

Finally, for every pair $f = \{x, y\}$ we trivially have

$$\bar{S}_T(f) \leq \left| \binom{V}{2} \right| = \binom{|V|}{2}, \tag{72}$$

hence

$$\bar{L}_T \leq \binom{|V|}{2}. \tag{73}$$

This completes the proof. $\qquad\square$

## B.3   Proof of Theorem 3

**Theorem 3** (Analysis of the Hamming–Lipschitz bound). *Let $d : \binom{V}{2} \to \mathbb{R}_{\geqslant 0}$ be a dissimilarity and let $T = (V, E(T))$ be the (tie-broken under Assumption 1) MST of $d$. For $e \in E(T)$ write its fundamental cut $C_e = (A_e, B_e)$ and $w_e = d(e)$. For an edited pair $f = \{x, y\}$ and edited dissimilarity $\tilde{d}$ with $\tilde{d}(g) = d(g)$ for $g \notin \{f\}$, define*

$$\Xi(f) := (\{f\} \cap E(T)) \cup \{\, e \in E(T) : f \text{ crosses } C_e \text{ and } \tilde{d}(f) < w_e \,\}, \qquad S_{\text{union}}(f) := \left| \bigcup_{e \in \Xi(f)} \mathcal{R}_e \right|. \tag{74}$$

*Then:*

> *(i)* (Tree-edge edit under strict cut separation) If $f = e \in E(T)$ and $e$ is strictly cut-separated, i.e. $\Delta_e(d) = w_e^+(d) - w_e > 0$, then there exists $\tilde{d}$ supported on $\{f\}$ such that
>
> $$\|u_d - u_{\tilde{d}}\|_0 = |A_e||B_e| = S_{\text{union}}(f). \tag{75}$$
>
> *(ii)* (Off-tree sharpness on an explicit family) There exist dissimilarities $d$, off-tree edges $f \notin E(T)$, and single-edge edits $\tilde{d}$ supported on $\{f\}$ such that
>
> $$\|u_d - u_{\tilde{d}}\|_0 = \left| \bigcup_{e \in \Xi(f)} \mathcal{R}_e \right| = S_{\text{union}}(f) = \Theta(n^2). \tag{76}$$
>
> Thus, the upper bound of Theorem 2 is attained on explicit off-tree instances, and in the worst case a single off-tree edit can force a quadratic number of ultrametric changes.
>
> *(iii)* (Necessity of instance dependence) Consequently, no universal subquadratic function $c(n) = o(n^2)$ can satisfy
>
> $$\|u_d - u_{\tilde{d}}\|_0 \le c(n) \|d - \tilde{d}\|_0 \tag{77}$$
>
> for all instances.

*Proof.* Throughout, recall from Lemma 1 the MST bottleneck representation $u_d(i,j) = \max_{e \in P_T(i,j)} w_e$ and from Lemma 2, the cut property argument: every $i$–$j$ path must cross every fundamental cut $C_e$ with $e \in P_T(i,j)$; if a crossing edge has weight $< w_e$, then the bottleneck along some $i$–$j$ path is $< w_e$.

*(i) Tree-edge edit.* Fix a tree edge $e \in E(T)$ that is strictly cut-separated, so that

$$\Delta_e(d) = w_e^+(d) - w_e > 0. \tag{78}$$

Choose $\varepsilon$ such that

$$0 < \varepsilon < \Delta_e(d), \tag{79}$$

and define $\tilde{d}(e) := w_e + \varepsilon$, while $\tilde{d}(g) := d(g)$ for all $g \neq e$. Consider any unordered pair $\{i,j\} \in \mathcal{R}_e$. Since $e \in P_T(i,j)$, the MST bottleneck representation from Lemma 1 gives

$$u_d(i,j) = w_e. \tag{80}$$

Under $\tilde{d}$, the tree path bottleneck becomes $w_e + \varepsilon$. Moreover, every other edge $g$ crossing the cut $C_e$ satisfies

$$d(g) \ge w_e^+(d) = w_e + \Delta_e(d) > w_e + \varepsilon, \tag{81}$$

so no alternative $i$–$j$ path can have bottleneck below $w_e + \varepsilon$. Hence

$$u_{\tilde{d}}(i,j) = w_e + \varepsilon > u_d(i,j), \tag{82}$$

and the pair changes.

If $\{i,j\} \notin \mathcal{R}_e$, then $e \notin P_T(i,j)$. No cut along $P_T(i,j)$ has acquired a strictly lighter crossing than its tree-edge weight, and none of the tree edges on $P_T(i,j)$ was edited. By the cut argument (as in the proof of Theorem 2), this implies $u_{\tilde{d}}(i,j) = u_d(i,j)$. Hence *exactly* the pairs in $\mathcal{R}_e$ change, so $\|u_d - u_{\tilde{d}}\|_0 = |A_e||B_e| = S_{\text{union}}(e)$.

*(ii) Off-tree sharpness on an explicit family.* Let $V = A \cup B$ with $|A| = \lfloor n/2 \rfloor$ and $|B| = \lceil n/2 \rceil$. Define $d$ by

$$d(i,j) = 1 \quad \text{if } \{i,j\} \subseteq A \text{ or } \{i,j\} \subseteq B, \tag{83}$$

and

$$d(i, j) = 10 \quad \text{if } i \in A, \ j \in B, \tag{84}$$

except for one distinguished cross edge $e^\star = \{a^\star, b^\star\}$ with

$$d(e^\star) = 8. \tag{85}$$

Then the unique MST $T$ consists of a unit-weight tree on $A$, a unit-weight tree on $B$, and the bridge $e^\star$ of weight 8.

Pick any other cross edge $f = \{a, b\} \in A \times B$, $f \neq e^\star$, and define $\tilde{d}$ by

$$\tilde{d}(f) = 2, \qquad \tilde{d}(g) = d(g) \text{ for } g \neq f. \tag{86}$$

We first identify the exposed-cut set $\Xi(f)$. Since $f \notin E(T)$, the first term in the definition vanishes. Along the tree path $P_T(a, b)$, every edge internal to $A$ or $B$ has weight 1, while the unique bridge edge $e^\star$ has weight 8. Because $\tilde{d}(f) = 2$, the edit does not expose any unit-weight tree edge, but it does expose the bridge cut:

$$2 \nless 1, \qquad 2 < 8. \tag{87}$$

Hence

$$\Xi(f) = \{e^\star\}, \qquad S_{\text{union}}(f) = |\mathcal{R}_{e^\star}| = |A||B|. \tag{88}$$

Now take any $i \in A$ and $j \in B$. Under $d$, every $i$–$j$ tree path must cross $e^\star$, so

$$u_d(i, j) = 8. \tag{89}$$

Under $\tilde{d}$, the path

$$i \rightsquigarrow a \ \rightarrow \ f \ \rightarrow \ b \rightsquigarrow j \tag{90}$$

has edge weights $1, 2, 1$, hence bottleneck 2. Therefore

$$u_{\tilde{d}}(i, j) \leq 2 < 8 = u_d(i, j), \tag{91}$$

so every cross pair changes. For pairs contained entirely inside $A$ or entirely inside $B$, the original unit-weight tree paths remain available and still have bottleneck 1, while the edited cross edge $f$ cannot produce a path of bottleneck below 1. Hence within-side pairs do not change.

Therefore the changed-pair set is exactly

$$\left\{ \{i, j\} \in \binom{V}{2} : i \in A, \ j \in B \right\}, \tag{92}$$

and

$$\|u_d - u_{\tilde{d}}\|_0 = |A||B| = S_{\text{union}}(f) = \Theta(n^2). \tag{93}$$

This proves exact attainability of the Theorem 2 bound on an explicit off-tree family.

$\square$

*(iii) Necessity of instance dependence.* This follows immediately from part (ii), which exhibits a single-edge edit with

$$\|d - \tilde{d}\|_0 = 1 \quad \text{and} \quad \|u_d - u_{\tilde{d}}\|_0 = \Theta(n^2). \tag{94}$$

## B.4    Proof of Corollary 1

**Corollary 1.** *Fix a minimum spanning tree $T$ of $d$, an edit set $F \subseteq \binom{V}{2}$, and edited weights $\tilde{d}$ supported on $F$. For each $f \in F$, define the exposed region $R(f) := \bigcup_{e \in \Xi(f)} \mathcal{R}_e$, so that $|R(f)| = S_{\text{union}}(f)$. Assume*

*that for each $f \in F$ there exists a certified changed-pair set $Q(f) \subseteq R(f)$ such that every pair in $Q(f)$ indeed changes under the common edited dissimilarity $\tilde{d}$, i.e. $Q(f) \subseteq \{\{i,j\} \in \binom{V}{2} : u_{\tilde{d}}(i,j) \neq u_d(i,j)\}$. Then:*

*(i)*

$$\left| \bigcup_{f \in F} Q(f) \right| \leq \|u_d - u_{\tilde{d}}\|_0 \leq \left| \bigcup_{f \in F} R(f) \right| \leq \sum_{f \in F} S_{\text{union}}(f).$$

*(ii) Moreover, consider any asymptotic regime of instances (for example, $|V| = n \to \infty$) in which*

$$\sum_{f \in F} |Q(f)| = (1 - o(1)) \sum_{f \in F} S_{\text{union}}(f),$$

*and the certified regions have asymptotically negligible total overlap:*

$$\sum_{\substack{f, f' \in F \\ f < f'}} |Q(f) \cap Q(f')| = o\left( \sum_{f \in F} |Q(f)| \right).$$

*Then*

$$\|u_d - u_{\tilde{d}}\|_0 = (1 - o(1)) \sum_{f \in F} S_{\text{union}}(f).$$

*Proof. (i)* Let

$$C := \big\{\{i,j\} : u_{\tilde{d}}(i,j) \neq u_d(i,j)\big\}$$

denote the set of unordered pairs whose ultrametric value changes.

We first prove the finite-sample sandwich bound.

By Theorem 2, every changed pair must lie in the exposed region of at least one edited edge. Therefore

$$C \subseteq \bigcup_{f \in F} R(f).$$

Taking cardinalities gives

$$|C| \leq \left| \bigcup_{f \in F} R(f) \right|.$$

Since the cardinality of a union is at most the sum of the cardinalities,

$$\left| \bigcup_{f \in F} R(f) \right| \leq \sum_{f \in F} |R(f)| = \sum_{f \in F} S_{\text{union}}(f).$$

Hence

$$|C| \leq \left| \bigcup_{f \in F} R(f) \right| \leq \sum_{f \in F} S_{\text{union}}(f).$$

*(ii)* On the other hand, by assumption, each certified set $Q(f)$ consists only of pairs that do change under the common edited dissimilarity $\tilde{d}$. Thus

$$Q(f) \subseteq C \qquad \text{for every } f \in F,$$

and therefore

$$\bigcup_{f \in F} Q(f) \subseteq C.$$

Taking cardinalities yields

$$\left| \bigcup_{f \in F} Q(f) \right| \leq |C|.$$

Combining the lower and upper bounds on $|C| = \|u_d - u_{\tilde{d}}\|_0$, we obtain

$$\left| \bigcup_{f \in F} Q(f) \right| \leq \|u_d - u_{\tilde{d}}\|_0 \leq \left| \bigcup_{f \in F} R(f) \right| \leq \sum_{f \in F} S_{\text{union}}(f).$$

This proves the first claim.

We now prove the asymptotic near-additivity statement. By the elementary first-order inclusion–exclusion bound,

$$\left| \bigcup_{f \in F} Q(f) \right| \geq \sum_{f \in F} |Q(f)| - \sum_{\substack{f, f' \in F \\ f < f'}} |Q(f) \cap Q(f')|.$$

Under the aggregate-overlap assumption

$$\sum_{\substack{f, f' \in F \\ f < f'}} |Q(f) \cap Q(f')| = o\left( \sum_{f \in F} |Q(f)| \right),$$

it follows that

$$\left| \bigcup_{f \in F} Q(f) \right| = (1 - o(1)) \sum_{f \in F} |Q(f)|.$$

Using the additional assumption

$$\sum_{f \in F} |Q(f)| = (1 - o(1)) \sum_{f \in F} S_{\text{union}}(f),$$

we conclude that

$$\left| \bigcup_{f \in F} Q(f) \right| = (1 - o(1)) \sum_{f \in F} S_{\text{union}}(f).$$

Finally, from the already established sandwich bound,

$$\left| \bigcup_{f \in F} Q(f) \right| \leq \|u_d - u_{\tilde{d}}\|_0 \leq \sum_{f \in F} S_{\text{union}}(f).$$

The lower bound is asymptotically $(1 - o(1)) \sum_{f \in F} S_{\text{union}}(f)$, while the upper bound is exactly $\sum_{f \in F} S_{\text{union}}(f)$. Therefore

$$\|u_d - u_{\tilde{d}}\|_0 = (1 - o(1)) \sum_{f \in F} S_{\text{union}}(f).$$

This proves the corollary. □

# C   Empirical Case Study: MST based Superpixel Segmentation

This appendix provides a complementary low-dimensional illustration of the structural score $S_{\text{union}}(e)$ in a downstream image-segmentation setting. Whereas the main-paper experiment in Section 4 directly measures ultrametric Hamming distortion on representation graphs, the present experiment studies whether the same score can identify *safe* versus *fragile* tree edges in a superpixel hierarchy.

**Motivation from the theory.**   Theorems 1 and 2 say that sparse edits propagate through the ultrametric only via edited or newly exposed MST cuts, and that for tree-edge edits the natural structural quantity is again

$$S_{\text{union}}(e) = |A_e|\,|B_e|. \tag{95}$$

In the segmentation setting, this suggests the following heuristic question: if one wants to perturb edges while causing as little downstream damage as possible, is it better to choose edges with small structural score than edges that merely have small local weight? The experiment below is designed to compare exactly those two notions of "safe" edits.

## C.1   Setup

We use the classical Cameraman image, shown in Figure 3. The grayscale image is converted to floating-point values in $[0,1]$ and oversegmented into $K$ SLIC superpixels. Each superpixel becomes a node in a region adjacency graph; neighboring superpixels are connected, and each edge is assigned a feature distance based on mean intensity and normalized centroid coordinates.

We then compute the tie-broken MST $T$ and the induced ultrametric $u_d$. A reference segmentation is obtained by cutting the $K-1$ heaviest MST edges and taking the resulting connected components; this is the standard MST view of single linkage. The resulting reference partitions for $K = 7, 8, 9, 10$ are shown in the top row of Figure 4.

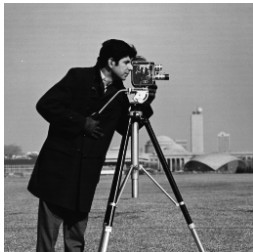

Figure 3: The classic Cameraman image.

For each tree edge $e = \{a, b\}$, we compute three quantities:

- the structural score

$$S_{\text{union}}(e) = |A_e|\,|B_e|; \tag{96}$$

- the raw edge weight $w(e)$, representing local boundary contrast;

- the worst-case segmentation impact

$$\text{impact}(e) := \max\Big\{1 - \text{ARI}\big(\text{base}, \text{decrease}(e)\big),\ 1 - \text{ARI}\big(\text{base}, \text{increase}(e)\big)\Big\}, \tag{97}$$

  where $\text{decrease}(e)$ and $\text{increase}(e)$ are the segmentations obtained after multiplying the weight of $e$ by $10^{-2}$ or $10^{2}$, respectively.

## C.2   Safe-edit curves

For each MST edge $e$, we therefore have a structural score $S_{\text{union}}(e)$, a local score $w(e)$, and an empirical damage score $\text{impact}(e)$. For each ranking rule ($S_{\text{union}}$ or $w$), we sort MST edges in ascending order, so that smaller score means "safer." For a prefix size $k$ (shown on the x-axis as a fraction of all MST edges), we take the bottom-$k$ edges under that ranking as the safe-to-edit set and report the maximum $\text{impact}(e)$ inside that set. This produces the two safe-edit curves shown in the bottom row of Figure 4.

The comparison is intentionally aligned with the theory. The score $w(e)$ is a purely local boundary heuristic, whereas $S_{\text{union}}(e)$ is derived from the exposed-cut geometry of Theorem 2. Thus, if the theory is capturing a meaningful notion of structural vulnerability, then ranking edges by $S_{\text{union}}(e)$ should produce safer edit sets than ranking them by raw weight alone.

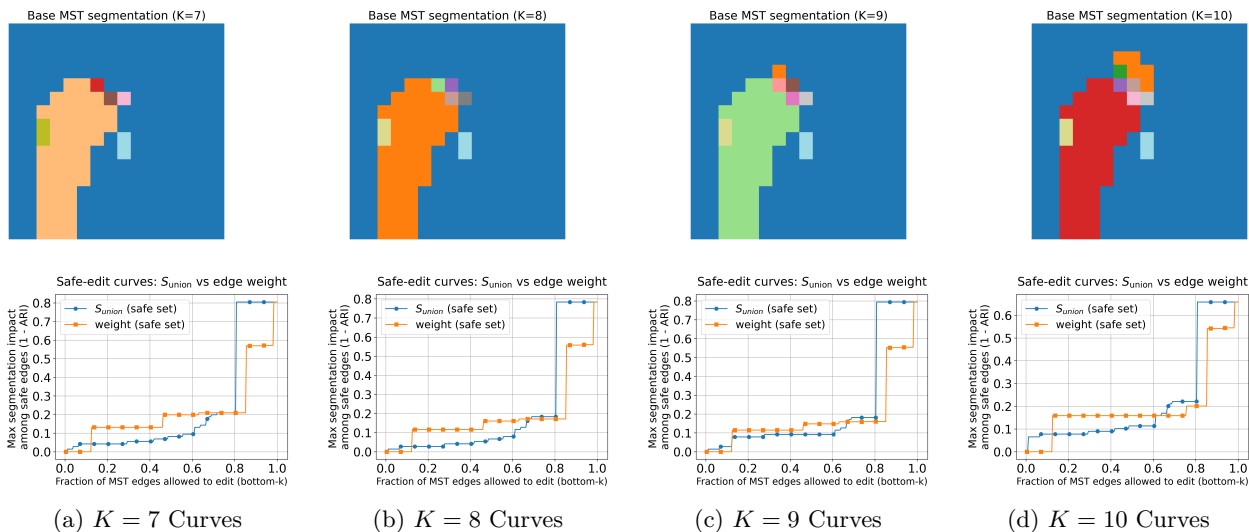

Figure 4: Safe-edit curves (bottom row) and corresponding reference segmentations (top row) for Cameraman image across $K = 7$–10. For each $K$, we report worst-case $1 - \text{ARI}$ when edits are restricted to bottom-$k$ MST edges ranked by $S_{\text{union}}(e)$ or by $w(e)$.

### C.3 Results

Figure 4 (bottom row) shows that, across $K = 7, 8, 9, 10$, the safe-edit curve based on $S_{\text{union}}(e)$ consistently lies below the curve based on $w(e)$. In other words, when the safe-edit budget is fixed, the worst damage incurred by editing low-$S_{\text{union}}$ edges is smaller than the worst damage incurred by editing edges selected solely by local boundary contrast. This indicates that the structural score derived from the theory yields a more reliable notion of "safe directions to perturb" than a purely local heuristic.

## D    Active MST-Edge Verification for Semi-Supervised Clustering

**Motivation.**   Our perturbation analysis identifies a simple structural quantity for a tree edge,

$$S_{\text{union}}(e) = |A_e|\,|B_e|,$$

where $A_e$ and $B_e$ are the two connected components obtained by deleting $e$ from the MST. This score measures how many cross-component pairs are exposed by a cut and therefore how many pairwise ultrametric relations can potentially change when that edge is corrected. This suggests a concrete human-in-the-loop use case for hierarchical clustering: if a practitioner can verify only a small number of MST edges using a human annotator, metadata source, or trusted secondary model, which edges should be checked first? We study this question as a budgeted *active MST-edge verification* problem and use it to evaluate whether the theory-derived score $S_{\text{union}}(e)$ yields a strong query policy in practice.

### D.1    Task and scope of the comparison

We consider a fixed-backbone active verification problem. Starting from a common feature representation, we build a single sparse graph, compute a single minimum spanning tree, and evaluate several edge-ranking rules on that same tree. Thus, the experiment is *not* a comparison of different end-to-end clustering pipelines or different graph-construction schemes. Instead, it asks a narrower and cleaner question:

> *Given a fixed sparse single-linkage backbone and a limited verification budget, which MST-edge priority rule most effectively improves the resulting clustering?*

This shared-backbone design is deliberate. If each baseline were allowed to build its own graph and its own tree, then differences in performance would conflate two effects: the quality of the backbone itself and the quality of the query rule imposed on that backbone. By holding the backbone fixed, we isolate the practical question relevant to a user who already has a hierarchical clustering and can only afford to verify a small number of edges.

### D.2 Datasets

The benchmark is run on the following full datasets:

- **MNIST** (LeCun et al., 2002),

- **USPS** (Liu et al., 2018),

- **HAR** (Anguita et al., 2013),

- **Olivetti Faces** (Credit to Cambridge University),

- **OptDigits** (Kaynak, 1995) .

These datasets span handwritten digits, human activity recognition, and face recognition, and therefore provide a heterogeneous test bed for active verification on MST-induced hierarchies. The goal of this suite is not to optimize performance for any one modality, but to test whether the query rules behave consistently across datasets with different geometry, class counts, and sample sizes.

### D.3 Common preprocessing pipeline

All methods share the same preprocessing pipeline. For each dataset, the raw features are:

1. standardized,

2. projected to 32 dimensions by PCA whenever the ambient dimension exceeds 32,

3. and then $\ell_2$-normalized.

This produces a common normalized feature space in which the sparse neighborhood graph and the MST are constructed.

Using a fixed preprocessing pipeline is important here. The aim is not to perform dataset-specific tuning for each baseline, but to compare edge-ranking policies under the same representation and the same hierarchical backbone.

### D.4 Sparse graph construction and common MST backbone

Let the preprocessed dataset be

$$\mathcal{X} = \{x_1, \ldots, x_n\}, \qquad y_i \in \{1, \ldots, K\}.$$

The labels are used only to simulate the oracle and to evaluate the final clustering; none of the query rules uses label information.

To scale the benchmark to full datasets, we do not construct the complete weighted graph. Instead, for each point we compute its $k$ nearest neighbors in feature space with

$$k = 100,$$

and retain only those sparse neighborhood edges. Let

$$d_{ij}^{\mathrm{raw}} = \|x_i - x_j\|_2$$

denote the raw Euclidean distance on the retained $k$-NN graph.

To reduce sensitivity to local density variation, we use a locally scaled metric. Let $\sigma_i$ be the distance from point $i$ to its $k_0$-th nearest neighbor in the raw neighborhood graph, with

$$k_0 = 7.$$

The locally scaled edge weight is then

$$d_{ij}^{\mathrm{ls}} = \frac{d_{ij}^{\mathrm{raw}}}{\sqrt{\sigma_i \sigma_j}}.$$

We compute the MST of this sparse graph under the locally scaled weights. The resulting tree is the common combinatorial backbone for all query policies.

This choice should be interpreted carefully. For example, the raw-weight baseline does *not* build a separate raw-distance MST. Rather, it ranks the edges of the common locally scaled MST by their raw Euclidean lengths. Likewise, the structural and feature-aware baselines are all evaluated on the same fixed tree. This keeps the candidate edge set identical across methods.

### D.5 Unverified baseline clustering

Before any active verification, we form an *unverified baseline clustering* by cutting the $(K-1)$ heaviest edges of the common MST under the backbone ordering. This produces exactly $K$ connected components and serves as the reference clustering against which all budgeted query policies are compared.

Formally, if the common MST edges are ordered by decreasing backbone weight as

$$e_{(1)}, e_{(2)}, \ldots, e_{(n-1)},$$

then the unverified baseline partition is obtained by deleting

$$\{e_{(1)}, \ldots, e_{(K-1)}\}.$$

### D.6 Active verification protocol

A query policy provides an ordering of MST edges. Given a query budget $b$, we inspect the first $b$ edges in that policy's ranking. For a queried edge

$$e = \{u, v\} \in E_T,$$

the oracle returns

$$e \text{ is wrong} \iff y_u \neq y_v.$$

Let

$$\mathcal{C}_b = \{e \in E_T : e \text{ was queried among the first } b \text{ edges and declared wrong}\}$$

denote the set of queried edges that are found to be incorrect.

The post-verification clustering is obtained in two stages:

1. every edge in $\mathcal{C}_b$ is forced to be cut;

2. if fewer than $(K-1)$ cuts have been made, we cut additional heaviest edges from the same common backbone ordering until the forest has exactly $K$ connected components.

Thus, all methods are compared under the same supervision budget, the same oracle, the same target number of clusters, and the same final backbone-completion rule. This protocol should therefore be viewed as a comparison of *which queried corrections are most useful within a shared sparse single-linkage correction pipeline*, not as a comparison of fully independent end-to-end algorithms.

This distinction matters. Because the final partition is completed using the common backbone order, the benchmark isolates the value of the *verified cuts selected by each policy* while keeping the downstream completion mechanism fixed. That makes the protocol practically relevant for settings where the backbone hierarchy is treated as given and only a limited number of corrections can be injected into it.

**Budgets.** Since an MST on $n$ vertices has

$$m = n - 1$$

edges, we use the budget set

$$b \in \{0, \lceil 0.005m \rceil, \lceil 0.01m \rceil, \lceil 0.02m \rceil, \lceil 0.05m \rceil, \lceil 0.10m \rceil\}.$$

Equivalently, we inspect approximately 0%, 0.5%, 1%, 2%, 5%, and 10% of the tree edges.

**Random baseline.** For the random policy, we select $b$ MST edges uniformly at random without replacement. Because this baseline is stochastic, we report the mean and standard deviation over 10 independent trials at each budget.

### D.7 Our method: the structural score $S_{\mathrm{union}}(e)$

For an MST edge $e \in E_T$, deleting $e$ partitions the tree into two connected components,

$$A_e \sqcup B_e = V.$$

Our theory-derived score is

$$S_{\mathrm{union}}(e) = |A_e|\,|B_e|.$$

**Interpretation.** This quantity counts the number of cross-component pairs exposed by cutting the edge. Every pair $(i, j)$ with $i \in A_e$ and $j \in B_e$ has its unique tree path crossing $e$, so any structural change at $e$ can potentially affect the ultrametric relation of all such pairs. Large values of $S_{\mathrm{union}}(e)$ therefore identify *load-bearing* edges: edges whose correction can affect a large portion of the induced hierarchy.

**Connection to the theory.** This is precisely the viewpoint suggested by the Hamming-stability analysis. Sparse perturbations do not propagate uniformly through the tree; their effect is mediated by the exposed cut-pair set. The quantity $|A_e||B_e|$ is exactly the size of that exposed set for a single edge. The resulting score is therefore not an ad hoc heuristic, but the operational form of the structural quantity singled out by the perturbation analysis.

**Query rule.** Our method ranks MST edges by decreasing $S_{\mathrm{union}}(e)$ and verifies them in that order.

### D.8 Benchmark baselines

We compare against the following baselines, all evaluated on the same common MST backbone.

#### D.8.1 Raw-weight baseline (Zahn, 2006)

For an MST edge

$$e = \{u, v\},$$

the raw-weight score is

$$w_{\mathrm{raw}}(e) = d_{uv}^{\mathrm{raw}}.$$

This baseline ranks edges by decreasing raw Euclidean length. It is the most direct local heuristic: long edges are treated as suspicious bridges. However, it uses only endpoint geometry and ignores the structural role of the edge inside the tree.

#### D.8.2 Scaled-weight baseline (Zelnik-Manor & Perona, 2004)

The scaled-weight baseline ranks edges by their backbone weight,

$$w_{\mathrm{ls}}(e) = d_{uv}^{\mathrm{ls}}.$$

Relative to raw edge length, this score discounts purely density-driven effects and therefore provides a stronger local geometric baseline on the shared sparse tree.

### D.8.3 Centroid-gap baseline (McQueen, 1967)

For an edge $e$, let $A_e$ and $B_e$ be the two connected components obtained by deleting $e$. Define their centroids by

$$\mu_{A_e} = \frac{1}{|A_e|} \sum_{i \in A_e} x_i, \qquad \mu_{B_e} = \frac{1}{|B_e|} \sum_{i \in B_e} x_i.$$

The centroid-gap score is

$$G_{\text{cg}}(e) = \|\mu_{A_e} - \mu_{B_e}\|_2^2.$$

This baseline ignores the local edge weight and instead asks whether the two sides of the cut are well separated in feature space.

### D.8.4 Ward-bridge baseline (Ward Jr, 1963)

The Ward-bridge score is

$$G_{\text{Ward}}(e) = \frac{|A_e||B_e|}{|A_e| + |B_e|} \|\mu_{A_e} - \mu_{B_e}\|_2^2.$$

This is the usual Ward merge penalty written as a split score on the tree. It combines centroid separation with component size and therefore acts as a stronger, size-aware feature baseline.

### D.8.5 Fisher-bridge baseline (Fisher, 1936)

Let

$$r(A_e) = \frac{1}{|A_e|} \sum_{i \in A_e} \|x_i - \mu_{A_e}\|_2^2, \qquad r(B_e) = \frac{1}{|B_e|} \sum_{i \in B_e} \|x_i - \mu_{B_e}\|_2^2$$

denote the mean squared within-component radii. The Fisher-bridge score is

$$G_{\text{Fisher}}(e) = \frac{\|\mu_{A_e} - \mu_{B_e}\|_2^2}{r(A_e) + r(B_e) + \varepsilon},$$

where $\varepsilon > 0$ is a small numerical constant for numerical stability. This baseline favors cuts whose two sides are well separated relative to their internal spread.

### D.8.6 Random baseline

Finally, we include a random-query baseline that selects edges uniformly at random. This provides a lower-bound reference and checks that any observed gains are due to meaningful prioritization rather than merely the presence of supervision.

### D.9 Evaluation metrics

After each budgeted verification step, we compare the resulting clustering against the ground-truth labels using four metrics.

**Cluster purity.** If the final clustering is

$$\mathcal{P} = \{C_1, \ldots, C_M\},$$

its purity is

$$\text{Purity}(\mathcal{P}, y) = \frac{1}{n} \sum_{m=1}^{M} \max_{c \in \{1,\ldots,K\}} \big|\{i \in C_m : \ y_i = c\}\big|.$$

Purity is easy to interpret, though it is relatively forgiving to fragmentation.

**Normalized Mutual Information (NMI).** We report the normalized mutual information between the recovered cluster labels and the ground-truth labels. NMI measures global agreement between the two partitions.

**Adjusted Rand Index (ARI).** We also report the adjusted Rand index, which evaluates pairwise agreement between the recovered clustering and the ground truth while correcting for chance.

**Verified wrong edges.** Finally, we report the number of queried edges that are actually incorrect,

$$W_b = \#\{e \text{ queried at budget } b : y_u \neq y_v\}.$$

This diagnostic is useful because a good query policy need not maximize only the raw count of wrong edges discovered; what matters is whether the queried wrong edges are consequential for the final hierarchy.

### D.10 Efficient computation of the benchmark scores

All methods share the same preprocessing, sparse graph construction, and MST computation stages. The main computational distinction lies in how the candidate edges are scored once the tree has been built.

**Structural score.** The score $S_{\text{union}}(e)$ depends only on subtree sizes. After rooting the MST once, all values $S_{\text{union}}(e)$ are obtained in a single tree pass, which costs

$$O(n),$$

followed by

$$O(n \log n)$$

to sort the edges.

**Raw-weight and scaled-weight baselines.** These baselines already have one scalar value per MST edge, so after the tree is built they require only sorting:

$$O(n \log n).$$

**Centroid-gap, Ward-bridge, and Fisher-bridge baselines.** These baselines require component-level feature statistics. In the implementation, subtree sizes, subtree sums, and subtree squared-norm sums are computed in one rooted-tree pass. This yields all required centroids and within-component radii for every edge in

$$O(nd),$$

after which sorting again costs

$$O(n \log n).$$

Hence the total post-MST complexity for these feature-aware baselines is

$$O(nd + n \log n).$$

### D.11 What this experiment tests

This benchmark separates several distinct notions of edge importance:

1. **local geometric salience**, captured by raw and scaled edge weights;

2. **pure structural exposure**, captured by $S_{\text{union}}(e)$;

3. **component-level separation**, captured by centroid-gap;

4. **size-aware feature separation**, captured by Ward-bridge;

5. **separation relative to internal spread**, captured by Fisher-bridge.

This makes the experiment substantially more informative than a simple comparison against raw edge length or random querying. If $S_{\text{union}}(e)$ outperforms the local geometric baselines, then the theory is identifying more than long-edge effects. If it remains competitive with the feature-aware baselines, then the perturbation-theoretic notion of a load-bearing edge is capturing a practically useful structural proxy for meaningful corrections in the hierarchy.

### D.12 Empirical goal

The empirical goal is therefore modest and well defined: to test whether the theory-derived score

$$S_{\text{union}}(e) = |A_e|\,|B_e|$$

provides an effective and computationally cheap priority rule for budgeted edge verification on a fixed sparse MST backbone. In this sense, the experiment serves as a practical validation of the structural quantity identified by the perturbation analysis.

### D.13 Empirical summary

Results are shown in Fig. 5. Across the benchmark suite, querying by $S_{\text{union}}(e)$ consistently improves clustering quality more rapidly than raw-weight, scaled-weight, and random querying, and remains competitive with the stronger component-level baselines such as centroid-gap, Fisher-bridge, and Ward-bridge. This is the practical machine-learning role of the theory: the Hamming-stability-derived structural score yields an efficient active-verification policy for hierarchical clustering on full real datasets.

An important qualitative takeaway is that the purely structural score $S_{\text{union}}(e)$ is often on par with the more feature-aware Ward-bridge baseline. This is notable because the two rules arise from different principles. Ward-bridge is a classical geometric criterion based on between-component separation and component size, whereas $S_{\text{union}}(e)$ emerges directly from our ultrametric perturbation analysis as the number of cross-component pairs exposed by a cut. Thus, the experiments suggest that the load-bearingness identified by the theory is not merely a graph-theoretic curiosity: it aligns closely with strong feature-aware notions of meaningful splits while remaining simpler, tree-only, and theoretically motivated.

Furthermore, comparing the clustering performance against the raw count of verified wrong edges (Fig. 5, bottom rows) reveals a critical dynamic. The local geometric baselines—raw and scaled weight—successfully identify the highest absolute number of incorrect edges, yet correcting them yields almost no improvement in global clustering quality. In contrast, $S_{\text{union}}(e)$ and Ward-bridge trigger massive gains despite discovering significantly fewer incorrect edges in total. This empirically validates the core thesis of our perturbation analysis: not all tree edges are structurally equal. Local heuristics waste the supervision budget snipping off isolated, structurally irrelevant outliers, whereas $S_{\text{union}}(e)$ successfully targets the load-bearing bridges whose correction resolves macroscopic ultrametric violations across large component

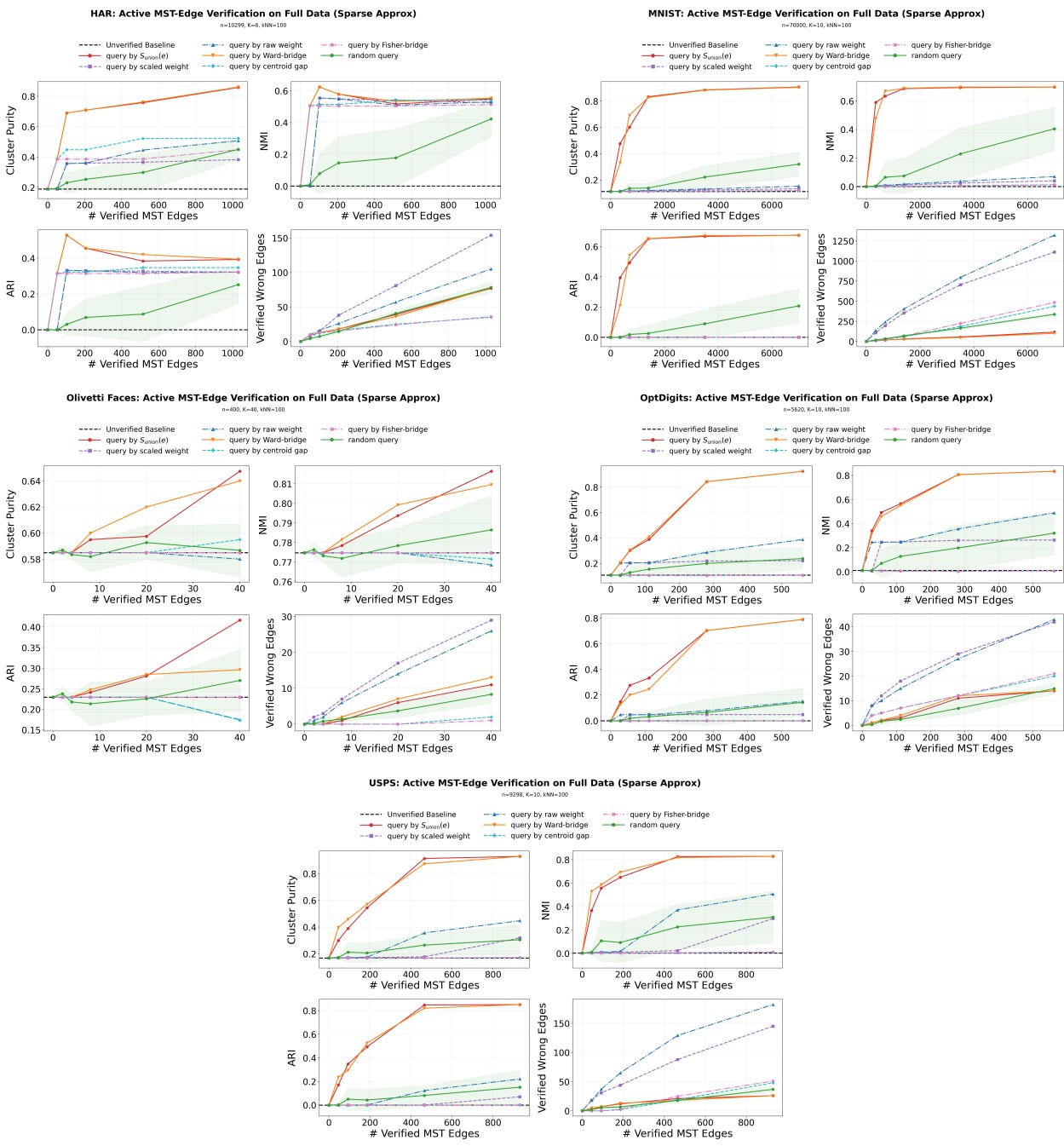

Figure 5: Budgeted active MST-edge verification for semi-supervised clustering. Clustering quality and verified-wrong-edge counts versus verification budget on a fixed sparse single-linkage backbone. The theory-derived score $S_{\text{union}}(e)$ provides an effective and computationally cheap query rule, outperforming raw/scaled-weight and random baselines and remaining competitive with stronger feature-aware policies.

# E   Substantiation of the Asymptotic Condition in Corollary 1(ii)

To substantiate the asymptotic condition in Corollary 1(ii), we construct a stylized "star of subtrees" regime. This demonstrates how simultaneous sparse edits can yield certified changed regions with asymptotically negligible aggregate overlap.

**Graph Construction.** Let the minimum spanning tree $T = (V, E(T))$ consist of a central hub node $v_0$ and $m$ distinct branches (subtrees) $B_1, \ldots, B_m$, each containing exactly $M$ nodes. The total number of nodes is $n = mM + 1$.

We assign the original dissimilarity $d$ as follows:

- $d(e) = 0$ for all internal subtree MST edges.

- $d(e_i) = 1$ for each of the $m$ hub-to-branch MST edges $e_i$.

- $d(e') = 3$ for every non-tree edge $e'$ crossing any branch cut.

This ensures each branch edge $e_i$ is strictly cut-separated, as the alternative crossing weight is strictly greater than the tree edge weight ($\Delta_{e_i} = 3 - 1 = 2 > 0$).

**Sparse Simultaneous Perturbation.** Let the perturbation set $F$ consist of $k$ distinct hub-to-branch edges, where $k = o(m)$. For each edited edge $f_i \in F$, the edited dissimilarity is strictly inflated to $\tilde{d}(f_i) = 2$. All other distances remain unchanged.

**Certification of Changed Regions.** For each edited branch edge $f_i$, its fundamental cut in $T$ separates the branch $B_i$ from the rest of the graph $V \setminus B_i$. The exposed cut-pair set is:

$$R(f_i) = \left\{ \{x, y\} \in \binom{V}{2} : x \in B_i, y \in V \setminus B_i \right\}.$$

By definition, the structural score is $S_{\text{union}}(f_i) = |R(f_i)| = M(n - M)$.

We must certify that every pair in $R(f_i)$ genuinely changes under the joint perturbation. Before the edits, every path from $B_i$ to $V \setminus B_i$ had an MST bottleneck of $u_d(x, y) = 1$. Under the joint perturbation $\tilde{d}$, the weight of $f_i$ increases to 2. Because every non-tree cross-edge has a weight of 3, no alternative path between $B_i$ and $V \setminus B_i$ can achieve a bottleneck below 2. Therefore, the ultrametric value strictly increases to $u_{\tilde{d}}(x, y) = 2$ for all $\{x, y\} \in R(f_i)$.

This guarantees that the certified changed region $Q(f_i)$ equals the entire exposed region:

$$|Q(f_i)| = |R(f_i)| = M(n - M).$$

**Asymptotic Aggregate Overlap.** For any pair of distinct edits $f_i, f_j \in F$ ($i \neq j$), the intersection $Q(f_i) \cap Q(f_j)$ consists exclusively of pairs that cross both fundamental cuts. In this topology, these are exactly the pairs with one endpoint in $B_i$ and the other in $B_j$. The size of this exact overlap is:

$$|Q(f_i) \cap Q(f_j)| = |B_i||B_j| = M^2.$$

Summing the overlap across all $\binom{k}{2}$ pairs of edits yields the aggregate overlap:

$$\sum_{i<j} |Q(f_i) \cap Q(f_j)| = \binom{k}{2} M^2 = \frac{k(k-1)}{2} M^2.$$

The sum of the individual certified regions is:

$$\sum_{i=1}^{k} |Q(f_i)| = kM(n - M) = kM(mM) = kmM^2.$$

Evaluating the ratio of the aggregate overlap to the total sum of the certified regions yields:

$$\frac{\sum_{i<j} |Q(f_i) \cap Q(f_j)|}{\sum_{i=1}^{k} |Q(f_i)|} = \frac{\frac{k(k-1)}{2} M^2}{kmM^2} = \frac{k-1}{2m}.$$

In the asymptotic regime where $k = o(m)$, we have:

$$\lim_{m \to \infty} \frac{k-1}{2m} = 0.$$

This confirms that the aggregate overlap is $o\left(\sum_{f \in F} |Q(f)|\right)$. Consequently, the conditions for Corollary 1(ii) are satisfied, and the total Hamming damage scales near-additively: $\|u_d - u_{\tilde{d}}\|_0 = (1 - o(1)) \sum_{f \in F} S_{\text{union}}(f)$.

