# OpenReview forum: "On Hamming–Lipschitz Type Stability of the Subdominant (Minmax) Ultrametric: Theory and Simple Proofs"
_TMLR — Accepted by TMLR_

### Review · Reviewer_qUAu · 2026-04-02

**Summary Of Contributions:**

The paper looks at the stability of the subdominant ultrametric of a given metric on a finite set of points. More precisely, the authors look at how much this ultrametric changes in Hamming distance (meaning the number of altered ultrametric values between pairs of points) when perturbing the original metric in the same fashion. The authors first relate this problem to the structure of the minimal spanning tree of the underlying graph. Then the authors provide a worst-case result showing that it can only make sense to look at this locally as on certain instances, the regularity constant scales quadratically with the number of points (which is also the combinatorial trivial upper bound for this constant). Finally, the authors provide elements of response (both theoretically and experimentally) on the prevalence of such catastrophic cases.

**Additional Comments:**

- Multiple objects and notations are not introduced in the right order. For instance, the term MST is used from the beginning of the article, but it is only defined as a short for Minimum Spanning Tree in Lemma 1. Similarly, the first paragraph of section 2 contains some graph-theory notations that are only defined later in the article. The same thing with "the tie-broken MST", it is used early in the article, without referencing the definition or justifying the uniqueness, and the definition and related properties only appear later in the article. Equation (11) uses notations that are only defined later in theorem 2. These inconsistencies make the paper read as if its structure was significantly altered without adapting the content.
- At page 7, the reference Cormel et al. does not feature a date.
- I do not understand the difference between definition 1 and definition 3; they seem to define the same objects with slightly different characterizations.
- Some objects are fixed in section 2 but are then redefined in every theorem, which breaks the flow.

**Audience:**

Yes

**Audience Explanation:**

I answered yes to the previous question, but it is a rather uncertain yes. While the authors do a great job contextualizing why minmax ultrametrics are useful in modern machine learning in section 1.2, the authors then state in the Takeaway paragraph that they are going to study a new property of those ultrametrics (namely the Hamming stability), but it is not clear that this property is actually useful in machine learning. Thus I am not sure if the contribution is interesting for TMLR's audience, or if it is a broader contribution in graph theory.

**Claims And Evidence:**

Yes

**Claims Explanation:**

While the overall writing could be improved, the article is theoretical in nature, and the authors provide full proofs of all of their results. The proofs appear to be sound.

**Requested Changes:**

- (critical) Can the authors detail more why the studied property is useful for machine learning?
- (critical) Can the authors fix the writing problems of the article that are listed in the "Additional Comments" section?
- (critical) Can the authors explain why the asymptotic condition expressed in (ii) of Corollary 1 is substantiated? For instance, by providing (even if toy) examples ?

---

> ### Author Response · Authors · 2026-04-20
> **Response to Reviewer qUAu x (1)**
>
> We are grateful to the reviewer for their evaluation and for the significant time and effort they invested in providing thoughtful and constructive feedback.
>
> $\textbf{(critical) Can the authors detail more why the studied property is useful for machine learning?}$
>
> $\textbf{Resp:}$ $\textbf{This response is split into multiple parts due to character limitations.}$
>
> We thank the reviewer for raising this important point. We completely agree that for a theory paper, the practical machine-learning applications must be front and center.
>
> Prompted by your feedback, we have significantly revised the manuscript to explicitly bridge the gap between our theoretical bounds and practical ML pipelines. First, we will clarify the concrete ML implications of our existing empirical work: the deep-embedding vulnerability study $\textbf{(Section 4)}$ and the downstream superpixel-segmentation case study $\textbf{(Appendix C)}$ by re-contextualizing to demonstrate how our theory translates into actionable structural diagnostics and safe-edit criteria.
>
> Furthermore, to provide an even stronger demonstration of downstream ML utility, we have added a new, semi-supervised clustering task in $\textbf{Appendix D}$. The key message unifying these experiments is that our theory is not just about whether the subdominant ultrametric is stable in an abstract sense; it provides a direct, computable method to identify which parts of a learned hierarchy are structurally fragile versus robust.
>
> $\textbf{(i)}$ First, the deep-embedding experiment in the main paper serves as our initial demonstration of this concrete ML utility. The paper shows that the score $S_{\mathrm{union}}(e)=|A_e||B_e|$, derived directly from $\textbf{Theorem 2}$, identifies a small high-risk tail of MST edges in DINO+UMAP representation graphs. Targeting those edges causes much larger normalized Hamming damage than random edits with the same budget, meaning that the score isolates a small number of load-bearing bridges whose perturbation can ripple widely through the induced hierarchy. In ML terms, this is a vulnerability map for hierarchical representations: it tells us where a representation is structurally brittle, rather than merely whether perturbations are large in norm.
>
> $\textbf{Use Case:}$ A concrete ML application is debugging embedding-based retrieval or clustering systems. For instance, in a medical-image or product-search pipeline built on DINO embeddings, a few spurious bridge edges may connect otherwise distinct semantic groups because of shortcut features such as background, texture, or imaging artifacts. Our score $S_{\mathrm{union}}(e)$ identifies exactly these high-impact connections, allowing practitioners to audit which edges most strongly distort the induced hierarchy. This is useful for targeted data cleaning, human review, and improving downstream clustering, retrieval, or pseudo-labeling reliability.
>
> $\textbf{(ii)}$ Second, to provide a clear example of downstream ML utility, we point to the superpixel-segmentation case study detailed in $\textbf{Appendix C}$. This appendix studies an MST-based superpixel segmentation task and asks a practically meaningful question: if one must perturb some edges, is it better to choose edges with low structural score rather than edges that merely have low local weight? The result is yes: restricting edits to low-$S_{\mathrm{union}}(e)$ edges preserves the downstream segmentation substantially better than heuristics based only on local edge weights. This is important because it moves the paper beyond “stress testing” and into a concrete downstream use case: the theory-derived score acts as a safe-edit criterion for a hierarchical segmentation pipeline. Appendix C was present in our previous manuscript's appendix for this reason, namely, to show that the theory has operational value in a downstream clustering/segmentation task, not only in synthetic perturbation analysis.
>
> $\textbf{Use case:}$ A concrete use case is human-in-the-loop correction of superpixel segmentations in settings such as tumor boundary refinement, road-network extraction from satellite images, or industrial defect inspection. In these pipelines, an operator often needs to edit a few graph edges to fix local mistakes. Our result says that low-$S_{\mathrm{union}}(e)$ edges are the ones that can be changed with the least downstream damage, so the score functions as a risk indicator for edits. This matters in practice because a wrong local edit can otherwise cascade into a much larger segmentation failure.
>
> $\textbf{To be contd...}$

---

> ### Author Response · Authors · 2026-04-20
> **Response to Reviewer qUAu x (2)**
>
> $\textbf{Continued..}$
>
> $\textbf{New Addition:}$ Third, to further strengthen the ML relevance, we added $\textbf{Appendix D}$. This appendix formulates a budgeted active MST-edge verification problem for semi-supervised clustering on full datasets including MNIST, USPS, HAR, Olivetti, and OptDigits. We ask the following question, “given a learned hierarchical backbone and limited supervision, which edges should be verified first?” In that setting, the theory-derived score $S_{\mathrm{union}}(e)$ serves as a computationally cheap priority rule, and the results show that it consistently improves clustering quality more rapidly than raw-weight, scaled-weight, and random querying, while remaining competitive with stronger feature-aware baselines such as centroid-gap, Fisher-bridge, and Ward-bridge. This directly demonstrates a machine-learning role for the theory: it yields an efficient supervision policy for correcting hierarchical cluster structures.
>
> $\textbf{Use case:}$ A concrete ML application is budgeted human verification in clustering pipelines. In settings such as handwritten-digit organization, activity discovery from wearable-sensor data, or face grouping in photo collections, only a small number of cluster connections can be checked by a human. Our score $S_{\mathrm{union}}(e)$ provides a cheap way to decide which edges to verify first so that limited supervision is spent on the connections most likely to improve the global cluster structure. Thus, the theory contributes an actionable supervision policy for semi-supervised clustering, not just a structural diagnostic.
>
> Hence, our theory gives concrete ML use case in budgeted human verification in graph-based clustering pipelines: when only a small number of pairwise links can be checked, $S_{\mathrm{union}}(e)$ derived from our $\textbf{Theorem 2}$ prioritizes the few MST edges whose correction can alter the largest part of the induced hierarchy, making it a practical query rule rather than merely a descriptive stability score.
>
> We also want to emphasize the theoretical side of the usefulness. In many ML settings that use single linkage or ultrametric projections, the relevant failure mode is not “all distances shift a little,” but rather “a few distances, affinities, or bridges are wrong.” Classical $\ell_\infty$, GH stability results control the magnitude of perturbation under uniform noise, but they do not control the extent of the induced structural damage when edits are sparse. Our contribution fills this gap precisely: $\textbf{Table 1}$ makes explicit that our Hamming-$\ell_0$​ theory is complementary to classical uniform stability, and $\textbf{Theorems 1–2}$ show that sparse perturbations propagate only through edited or newly exposed MST cuts. This is precisely the type of information one needs to reason about robustness, diagnosis, and intervention in hierarchical ML pipelines.
>
> So from an ML standpoint, the paper provides not just a new bound, but a structural language for identifying load-bearing cuts, prioritizing corrections, and distinguishing benign from high-impact perturbations in learned hierarchies.

---

> ### Author Response · Authors · 2026-04-20
> **Response to Reviewer qUAu x (3)**
>
> $\textbf{(critical) Can the authors explain why the asymptotic condition expressed in (ii) of Corollary 1 is substantiated?}$
> $\textbf{For instance, by providing (even if toy) examples ?}$
>
> $\textbf{Resp:}$ We thank the reviewer for this question. We agree that Corollary 1(ii) is conditional, and that the draft would benefit from an explicit family showing that its hypotheses are non-vacuous. We have included $\textbf{Appendix E}$ in the revised manuscript and provide a brief overview of its contents below. We respectfully invite the reviewer to examine the full section for a more detailed discussion.
>
> To substantiate the asymptotic condition in Corollary 1(ii), we can construct a natural, stylized regime, a $\textit{star of subtrees}$, which demonstrates how simultaneous sparse edits can yield certified changed regions with asymptotically negligible overlap.
>
> Below is a toy example:
>
> $\textbf{The Construction:}$
>
> Let the MST $T$ consist of a central hub node and $m$ distinct branches (subtrees), each containing exactly $M$ nodes.
>
> $\textbf{(i)}$ Assign weight 0 to all internal subtree MST edges.
>
> $\textbf{(ii)}$ Assign weight 1 to each of the $m$ hub-to-branch MST edges.
>
> $\textbf{(iii)}$ Assign weight 3 to every non-tree edge crossing any branch cut, ensuring each branch edge is strictly cut-separated.
>
> $\textbf{The Perturbation:}
>
> Consider a sparse, simultaneous edit of $k$ distinct hub-to-branch edges, where we increase their weights from 1 to 2. We assume the number of edits $k$ is small relative to the total number of branches, i.e., $k = o(m)$.
>
> $\textbf{Certification of Changed Regions:}$
>
> For each edited branch edge $f_i$, the exposed region $R(f_i)$ consists of all pairs separating subtree $i$ from the rest of the graph. Every path from subtree $i$ to the rest of the graph must cross this branch cut.
>
> Before the perturbation, the bottleneck of these paths was 1. Under the joint perturbation, the edited edge $f_i$ increases to 2. Crucially, because all non-tree crossings have a weight of 3, no alternative crossing below 2 exists. Therefore, every single pair in $R(f_i)$ genuinely changes its ultrametric value under the common edited dissimilarity.
>
> This certifies that the changed region equals the exposed region:
>
> $$|Q(f_i)| = |R(f_i)| = S_{union}(f_i) = M(n-M)$$
>
> $\textbf{Asymptotic Overlap:}$
>
> For any two distinct edits $f_i$ and $f_j$ (where $i \neq j$), the intersection $|Q(f_i) \cap Q(f_j)|$ consists exactly of the pairs with one endpoint in subtree $i$ and the other in subtree $j$. Therefore:
>
> $$|Q(f_i) \cap Q(f_j)| = M^2$$
>
> Summing the overlap across all pairs of the $k$ edits, and comparing it to the total sum of certified pairs, we get:
>
> $$\sum_{i<j} |Q(f_i) \cap Q(f_j)| \asymp \binom{k}{2}M^2$$
>
> $$\sum_i |Q(f_i)| \asymp k M(n-M) \asymp k m M^2$$
>
> Evaluating the ratio of the aggregate overlap to the sum of the individual regions yields:
>
> $$\frac{\sum_{i<j} |Q(f_i) \cap Q(f_j)|}{\sum_i |Q(f_i)|} \asymp \frac{k}{m} \to 0$$
>
> Whenever $k = o(m)$, this ratio vanishes. This exactly satisfies the asymptotic condition required by Corollary 1(ii), proving that in a topology where sparse edits afflict distinct semantic regions, the ultrametric damage scales near-additively without saturating.

---

> ### Author Response · Authors · 2026-04-20
> **Response to Reviewer qUAu x (4)**
>
> $\textbf{(critical) Can the authors fix the writing problems of the article that are listed in the "Additional Comments" section?}$
>
> $\textbf{NOTE:}$ All the changes below are reflected in colour red in the revised manuscript.
>
> $\textbf{(i)}$ Multiple objects and notations are not introduced in the right order. For instance, the term MST is used from the beginning of the article, but it is only defined as a short for Minimum Spanning Tree in Lemma 1. Similarly, the first paragraph of section 2 contains some graph-theory notations that are only defined later in the article. The same thing with "the tie-broken MST", it is used early in the article, without referencing the definition or justifying the uniqueness, and the definition and related properties only appear later in the article. Equation (11) uses notations that are only defined later in theorem 2. These inconsistencies make the paper read as if its structure was significantly altered without adapting the content.
>
> $\textbf{Resp:}$ We are deeply grateful to the reviewers for highlighting these inconsistencies. We have updated the abstract to include the MST terminology and thoroughly revised Section 2 to ensure the notation is consistent and follows a logical progression. The concept of a $\textit{tie-broken-MST}$ is now clearly defined and referenced wherever appropriate. Additionally, we have moved the definition of the notation in Eq. (11) earlier in the text for better clarity. Please let us know if any other points require attention; we are happy to address them immediately.
>
> $\textbf{(ii)}$  At page 7, the reference Cormel et al. does not feature a date
>
> $\textbf{Resp:}$ We have fixed it in our revised manuscript.
>
> $\textbf{(iii)}$ I do not understand the difference between definition 1 and definition 3; they seem to define the same objects with slightly different characterizations.
>
> $\textbf{Resp:}$ We appreciate the reviewer for pointing this out. Following your suggestion, we have merged these into a single, unified Definition 1.
>
> $\textbf{(iv)}$ Some objects are fixed in section 2 but are then redefined in every theorem, which breaks the flow.
>
> $\textbf{Resp:}$ We have eliminated the redundant use of $\Xi$ and $C_e$ and streamlined the presentation of our theorems for better conciseness. Please let us know if any further refinements are required; we would be happy to make additional adjustments.

---

### Review · Reviewer_R2cm · 2026-04-13

**Summary Of Contributions:**

The paper studies the stability of the subdominant (minmax) ultrametric under sparse perturbations. It introduces an l0 (Hamming-type) perspective, showing that changes propagate through MST cuts, and derives (1) localization results, (2) a per-edit exposed-cut score, and (3) a global Hamming–Lipschitz bound. It further establishes sharpness results and provides diagnostic experiments on learned embeddings.

**Audience:**

Yes

**Audience Explanation:**

The work addresses stability of hierarchical clustering from a sparsity-aware perspective, which is relevant to researchers in clustering, metric geometry, and representation learning. The l0 viewpoint complements existing l-infty/GH analyses and may be of interest to a portion of the TMLR audience.

**Broader Impact Concerns:**

The discussion is generally sufficient.

**Claims And Evidence:**

Yes

**Claims Explanation:**

The theoretical claims are supported by clear definitions, formal statements, and proofs with matching sharpness results. The arguments are internally consistent and align with known MST properties. The empirical section, while limited in scope, provides illustrative evidence consistent with the theoretical findings.

**Requested Changes:**

Not critical for acceptance but would strengthen the work:
Clarify practical implications of the l0 stability bounds, especially for downstream ML tasks or robustness analysis.
Provide additional intuition or simple examples illustrating exposed cuts and propagation behavior.
Discuss scenarios where the bounds may be loose or less informative (e.g., worst-case quadratic changes).

---

> ### Author Response · Authors · 2026-04-20
> **Response to Reviewer R2cm x (1)**
>
> We are grateful to the reviewer for their evaluation and for the significant time and effort they invested in providing thoughtful and constructive feedback.
>
> $\textbf{Clarify practical implications of the l0 stability bounds, especially for downstream ML tasks or robustness analysis}$
>
> $\textbf{Resp:}$ $\textbf{This response is split into multiple parts due to character limitations.}$
>
> We thank the reviewer for raising this important point. We completely agree that for a theory paper, the practical machine-learning applications must be front and center.
>
> Prompted by your feedback, we have significantly revised the manuscript to explicitly bridge the gap between our theoretical bounds and practical ML pipelines. First, we will clarify the concrete ML implications of our existing empirical work: the deep-embedding vulnerability study $\textbf{(Section 4)}$ and the downstream superpixel-segmentation case study $\textbf{(Appendix C)}$ by re-contextualizing to demonstrate how our theory translates into actionable structural diagnostics and safe-edit criteria.
>
> Furthermore, to provide an even stronger demonstration of downstream ML utility, we have added a new, semi-supervised clustering task in $\textbf{Appendix D}$. The key message unifying these experiments is that our theory is not just about whether the subdominant ultrametric is stable in an abstract sense; it provides a direct, computable method to identify which parts of a learned hierarchy are structurally fragile versus robust.
>
> $\textbf{(i)}$ First, the deep-embedding experiment in the main paper serves as our initial demonstration of this concrete ML utility. The paper shows that the score $S_{\mathrm{union}}(e)=|A_e||B_e|$, derived directly from $\textbf{Theorem 2}$, identifies a small high-risk tail of MST edges in DINO+UMAP representation graphs. Targeting those edges causes much larger normalized Hamming damage than random edits with the same budget, meaning that the score isolates a small number of load-bearing bridges whose perturbation can ripple widely through the induced hierarchy. In ML terms, this is a vulnerability map for hierarchical representations: it tells us where a representation is structurally brittle, rather than merely whether perturbations are large in norm.
>
> $\textbf{Use Case:}$ A concrete ML application is debugging embedding-based retrieval or clustering systems. For instance, in a medical-image or product-search pipeline built on DINO embeddings, a few spurious bridge edges may connect otherwise distinct semantic groups because of shortcut features such as background, texture, or imaging artifacts. Our score $S_{\mathrm{union}}(e)$ identifies exactly these high-impact connections, allowing practitioners to audit which edges most strongly distort the induced hierarchy. This is useful for targeted data cleaning, human review, and improving downstream clustering, retrieval, or pseudo-labeling reliability.
>
> $\textbf{(ii)}$ Second, to provide a clear example of downstream ML utility, we point to the superpixel-segmentation case study detailed in $\textbf{Appendix C}$. This appendix studies an MST-based superpixel segmentation task and asks a practically meaningful question: if one must perturb some edges, is it better to choose edges with low structural score rather than edges that merely have low local weight? The result is yes: restricting edits to low-$S_{\mathrm{union}}(e)$ edges preserves the downstream segmentation substantially better than heuristics based only on local edge weights. This is important because it moves the paper beyond “stress testing” and into a concrete downstream use case: the theory-derived score acts as a safe-edit criterion for a hierarchical segmentation pipeline. Appendix C was present in our previous manuscript's appendix for this reason, namely, to show that the theory has operational value in a downstream clustering/segmentation task, not only in synthetic perturbation analysis.
>
> $\textbf{Use case:}$ A concrete use case is human-in-the-loop correction of superpixel segmentations in settings such as tumor boundary refinement, road-network extraction from satellite images, or industrial defect inspection. In these pipelines, an operator often needs to edit a few graph edges to fix local mistakes. Our result says that low-$S_{\mathrm{union}}(e)$ edges are the ones that can be changed with the least downstream damage, so the score functions as a risk indicator for edits. This matters in practice because a wrong local edit can otherwise cascade into a much larger segmentation failure.
>
> $\textbf{To be contd...}$

---

> ### Author Response · Authors · 2026-04-20
> **Response to Reviewer R2cm x (2)**
>
> $\textbf{Continued..}$
>
> $\textbf{New Addition:}$ Third, to further strengthen the ML relevance, we added $\textbf{Appendix D}$. This appendix formulates a budgeted active MST-edge verification problem for semi-supervised clustering on full datasets including MNIST, USPS, HAR, Olivetti, and OptDigits. We ask the following question, “given a learned hierarchical backbone and limited supervision, which edges should be verified first?” In that setting, the theory-derived score $S_{\mathrm{union}}(e)$ serves as a computationally cheap priority rule, and the results show that it consistently improves clustering quality more rapidly than raw-weight, scaled-weight, and random querying, while remaining competitive with stronger feature-aware baselines such as centroid-gap, Fisher-bridge, and Ward-bridge. This directly demonstrates a machine-learning role for the theory: it yields an efficient supervision policy for correcting hierarchical cluster structures.
>
> $\textbf{Use case:}$ A concrete ML application is budgeted human verification in clustering pipelines. In settings such as handwritten-digit organization, activity discovery from wearable-sensor data, or face grouping in photo collections, only a small number of cluster connections can be checked by a human. Our score $S_{\mathrm{union}}(e)$ provides a cheap way to decide which edges to verify first so that limited supervision is spent on the connections most likely to improve the global cluster structure. Thus, the theory contributes an actionable supervision policy for semi-supervised clustering, not just a structural diagnostic.
>
> Hence, our theory gives concrete ML use case in budgeted human verification in graph-based clustering pipelines: when only a small number of pairwise links can be checked, $S_{\mathrm{union}}(e)$ derived from our $\textbf{Theorem 2}$ prioritizes the few MST edges whose correction can alter the largest part of the induced hierarchy, making it a practical query rule rather than merely a descriptive stability score.
>
> We also want to emphasize the theoretical side of the usefulness. In many ML settings that use single linkage or ultrametric projections, the relevant failure mode is not “all distances shift a little,” but rather “a few distances, affinities, or bridges are wrong.” Classical $\ell_\infty$, GH stability results control the magnitude of perturbation under uniform noise, but they do not control the extent of the induced structural damage when edits are sparse. Our contribution fills this gap precisely: $\textbf{Table 1}$ makes explicit that our Hamming-$\ell_0$​ theory is complementary to classical uniform stability, and $\textbf{Theorems 1–2}$ show that sparse perturbations propagate only through edited or newly exposed MST cuts. This is precisely the type of information one needs to reason about robustness, diagnosis, and intervention in hierarchical ML pipelines.
>
> So from an ML standpoint, the paper provides not just a new bound, but a structural language for identifying load-bearing cuts, prioritizing corrections, and distinguishing benign from high-impact perturbations in learned hierarchies.

---

> ### Author Response · Authors · 2026-04-20
> **Response to Reviewer R2cm x (3)**
>
> $Q : \textbf{Provide additional intuition or simple examples illustrating exposed cuts and propagation behavior.}$
> $\textbf{Discuss scenarios where the bounds may be loose or less informative (e.g., worst-case quadratic changes).}$
>
> $\textbf{Resp:}$ We thank the reviewer for highlighting this topic. While Theorem 3 proves our bounds are tight in the worst-case, there is a very common structural scenario where our Hamming-Lipschitz bound heavily overestimates the number of changed entries.  Our upper bound operates by counting all pairs that cross a newly "exposed cut." It assumes that if a local bridge gets cheaper, every pair relying on that bridge will see their overall ultrametric distance drop. However, this bound becomes loose when a pair's path contains a much larger bottleneck elsewhere in the tree.
>
>
> $\textbf{A Simple, Intuitive Example:}$
>
> Imagine a line graph with four nodes: A, B, C, and D.
>
> The MST edges are A-B (weight 10), B-C (weight 10), and C-D (weight 100).
>
> For the pair (A, D), the path is A-B-C-D. The maximum edge on this path is C-D (100). Therefore, the original ultrametric distance for (A, D) is 100.
>
> $\textbf{The Sparse Edit:}$ An adversary adds a new "shortcut" off-tree edge directly between A and C with a weight of 2.
> Because 2 is much cheaper than 10, this edit successfully "exposes" the A-B and B-C cuts. According to our theorem, the pair (A, D) crosses these exposed cuts, so it is included in the theoretical upper bound of pairs that might change.
>
>
> While the local route from A to C got much cheaper (dropping from 10 to 2), the pair (A, D) still must traverse the C-D edge (weight 100) to reach its destination. Their new path bottleneck is max(2, 100) = 100. Their ultrametric distance remains exactly 100.
>
> In this scenario, the bound counts pairs that experience no actual change, making it a loose overestimate. However, this looseness is an intentional theoretical trade-off. To calculate the exact number of changes, one would have to perform a global, O(n²) search to check every single pair's distant bottlenecks.
>
> By accepting this looseness, our exposed-cut score isolates the local structural risk of an edge without requiring global computation. As demonstrated in our empirical case studies, this purely structural, albeit sometimes loose, upper bound is highly effective at identifying the load-bearing vulnerabilities in real-world deep embedding graphs.

---

### Decision · Action_Editor_Trgi · 2026-06-24

**Recommendation:** Accept as is

**Additional Comments:**

The authors may want to combine their supplementary material and main paper for ease of reading.

**Audience:**

Yes

**Audience Explanation:**

The authors have taken care to provide examples of ML problems where their results may be relevant. TMLR's machine learning centric audience can appreciate this contribution with those examples as signposts.

**Claims And Evidence:**

Yes

**Claims Explanation:**

The authors derive a hamming-lipschitz stability bound for ultrametrics that capture the effects of sparse edits. The theoretical results seem correct and provide novel insights into metrics that can identify critical nodes in hierarchical clustering settings. The authors have cleaned up notational and presentation issues during the rebuttal process.

---

> ### Author Response · Authors · 2026-06-27
> **Camera Ready**
>
> Thank you for the comments and the acceptance. We have uploaded the camera-ready version.